# The spatial allocation of population: A review of large-scale gridded population data products and their fitness for use

Stefan Leyk[1*], Andrea E. Gaughan[2, 10], Susana B. Adamo[3], Alex de Sherbinin[3], Deborah Balk[9], Sergio Freire[5], Amy Rose[4], Forrest R. Stevens[2], Brian Blankespoor[8], Charlie Frye[7], Joshua Comenetz[6], Alessandro Sorichetta[10], Kytt MacManus[3], Linda Pistolesi[3], Marc Levy[3], Andrew J Tatem[10], Martino Pesaresi[5]

[1]Department of Geography, University of Colorado Boulder, Boulder, CO 80309, U.S.A.
[2]Department of Geography and Geosciences, University of Louisville, KY, 40292, U.S.A.
[3]CIESIN, Columbia University, Palisades, NY, 10964, U.S.A.
[4]Human Dynamics Group, Oak Ridge National Laboratory, Oak Ridge, TN, 37831, U.S.A.
[5]European Commission, Joint Research Centre (JRC), Ispra, Italy
[6]U.S. Census Bureau, Washington, D.C., 20233, U.S.A.
[7]Environmental Systems Research Institute, Redlands, CA 92373
[8]Development Research Group, World Bank, Washington, D.C. 20433, U.S.A.
[9]CUNY Institute for Demographic Research, and Marxe School of Public and International Affairs, Baruch College, City University of New York, 10010, USA
[10]WorldPop, School of Geography and Environmental Sciences, University of Southampton, Southampton, SO17 1BJ, UK

[*] *Correspondence to*: Stefan Leyk (stefan.leyk@colorado.edu)

**Abstract.** Population data represent an essential component in studies focusing on human-nature interrelationships, disaster risk assessment and environmental health. Several recent efforts have produced global and continental-extent gridded population data which are becoming increasingly popular among various research communities. However, these data products, which are of very different characteristics and based on different modeling assumptions, have never been systematically reviewed and compared which may impede their appropriate use. This article fills this gap and presents, compares and discusses a set of large-scale (global and continental) gridded datasets representing population counts or densities. It focuses on data properties, methodological approaches and relative quality aspects that are important to fully understand the characteristics of the data with regard to the intended uses. Written by the data producers and members of the user community, through the lens of the "fitness for use" concept, the aim of this paper is to provide potential data users with the knowledge base needed to make informed decisions about the appropriateness of the data products available in relation to the target application and for critical analysis.

**Short summary**: Population data are essential for studies on human-nature relationships, disaster or environmental health. Several global and continental gridded population data have been produced but have never been systematically compared. This article fills this gap and critically compares these gridded population datasets. Through the lens of the "fitness for use" concept it provides users with the knowledge needed to make informed decisions about appropriate data use in relation to the target application.

**Glossary of key terms:**

**Population grid**: A spatial representation of either population counts or population density within a number of quadrilateral grid cells, covering a given extent on the Earth surface.

**Spatial Resolution**: The size of a grid cell used to represent the cell value (also called granularity).

**Analytical scale**: The spatial scale (or level of aggregation) at which a given spatial analysis will be performed; related to spatial resolution.

**Precision**: The degree of exactness of a measurement.

**Accuracy**: Refers to how close the measurements are to the true values.

**Temporal Resolution**: The amount of time between two representations of the data covering the same area. For remotely sensed data, it depends on the time a sensor revisits and acquires data for the exact same location.

**Currency**: The temporal proximity of the data of interest to a given point in time.

**De facto population**: The number of persons who are physically present in a geographical area at the time of the enumeration.

**De jure population**: The number of persons attributed to a geographical area based on their legal or usual place of residence - regardless if they are present at the time of the enumeration.

**Areal interpolation**: The process of making estimates for a set of spatial units based on another incongruent set of spatial units that can be partially or entirely overlapping.

**Dasymetric mapping**: The process of spatially redistributing quantities through areal interpolation using ancillary data associated with the variable of interest.

**Fitness for use**: A concept to assess the characteristics and the level of relative quality/accuracy of a given dataset in relation to a given purpose or to fulfill the user needs.

**Modifiable Areal Unit Problem (MAUP)**: A source of statistical bias due to arbitrary spatial aggregation of data potentially resulting in non-representative results if the process of interest operates at different scales.

**Attribution Table.** Gridded population data collections described in this review article, years covered, digital object identifiers and reference links. This review covers sources and versions available as of May 2019.

| Data collection | Year(s) | Population Themes | Digital Object Identifier (doi) | Reference Link |
|---|---|---|---|---|
| Gridded Population of the World (GPWv4.11) | 2000; 2005; 2010; 2015; 2020 | Persons | 10.7927/H4JW8BX5 | http://sedac.ciesin.columbia.edu/data/collection/gpw-v4 |
| | | UN WPP-adj. | 10.7927/H4PN93PB | |
| | | Pop. Density | 10.7927/H49C6VHW | |
| | | UN WPP-adj. | 10.7927/H4F47M65 | |
| Global Rural Urban Mapping Project (GRUMPv1) | 1990; 1995; 2000 | Persons | 10.7927/H4VT1Q1H | http://sedac.ciesin.columbia.edu/data/collection/grump-v1 |
| | | Pop. Density | 10.7927/H4R20Z93 | |
| LandScan Global Population Database (Landscan Global) | annual: 2000–2016 | Persons | N/A; data download at: | https://landscan.ornl.gov/ |
| WorldPop | 2000-2020 | Persons | 10.5258/SOTON/WP00645 | www.worldpop.org |
| Global Human Settlement Layer - Population (GHS-POP) | 1975; 1990; 2000; 2015 | Persons | http://data.europa.eu/89h/jrc-ghsl-ghs_pop_gpw4_globe_r2015a | http://ghsl.jrc.ec.europa.eu/ghs_pop.php |
| World Population Estimate (WPE) | 2013 | Persons | 10.13140/RG.2.2.18213.14565 | https://sites.google.com/ciesin.columbia.edu/popgrid/find-data/esri |
| | 2015 | Persons Pop. Density | 10.13140/RG.2.2.16160.79367 10.13140/RG.2.2.14857.70248 | |
| | 2016 | Persons Pop. Density | 10.13140/RG.2.2.12996.48007 10.13140/RG.2.2.21568.58885 | |
| History Database of the Global Environment (HYDE) Population Grids v3.2 | 10,000 BC - 2015 | Persons | 10.17026/dans-25g-gez3 | https://themasites.pbl.nl/tridion/en/themasites/hyde/download/index-2.html |
| High Resolution Settlement Layer (HRSL) | 2015 | Persons | N/A; data download at | https://ciesin.columbia.edu/data/hrsl/ |
| European GHS Population Grid (GHS-POP-EUROSTAT) | 2011 | Persons | http://data.europa.eu/89h/jrc-ghsl-ghs_pop_eurostat_europe_r2016a | http://data.jrc.ec.europa.eu/dataset/jrc-ghsl-ghs_pop_eurostat_europe_r2016a |
| Gridded Population Mapping (Demobase) | 1998-present | Persons | N/A; data download at: | https://www.census.gov/geographies/mapping-files/time-series/demo/international-programs/demobase.html |

**1 Introduction**

The distribution and density of human population continues to be a critical component to measuring, mapping and understanding human-environment interrelationships, identifying populations at risk of infectious diseases or disasters, and informing management and policy decisions from local to global level initiatives (e.g. Wesolowski et al. 2014, Simarro et al. 2011, McDonald et al. 2011, Jones et al. 2008, McGranahan et al. 2007, Doocy et al. 2007). The traditional form of collecting population data is through a census or registry, and those population counts can be spatially linked to boundary datasets representing enumeration areas (the most basic unit of collected census data) or administrative units in a Geographic Information System (GIS). More recently, an increasing use of fully georeferenced censuses has made building-level mapping more feasible in some countries. However, census data vary substantially across countries with regard to quality, the number and size of enumerated areas, the frequency of data collection and the level of confidentiality depending on detail. The size of census units also varies significantly within countries between rural and urban areas. Thus, to be useful for many analytical purposes, substantial efforts are required to harmonize such enumerated data (de Sherbinin 2017, Zoraghein et al. 2016, Schroeder 2007). Since Tobler's "World population in a grid of spherical quadrilaterals" (Tobler et al. 1997) and Liverman et al.'s "People and Pixels" (Liverman et al. 1998), the benefits of gridded population data have been acknowledged. As a consequence, the scientific community has increasingly invested in ways to create global georeferenced data products that help overcome the inconsistencies in census-derived national population data and facilitate their integration with other gridded spatial datasets such as, for example, remote sensing data products. This article, a product of the POPGRID Data Collaborative (POPGRID 2018), describes the variety of gridded population data products that have been created over the past 20 years and is an effort to aid users in better understanding the nature of these products, their qualities and forms of appropriate uses.

There is high demand for modeled gridded population datasets particularly in countries with less detailed or infrequent censuses. These datasets, for example, support land use and urban planning (Dong et al. 2017), measurement of economic development (Nordhaus 2006, Uchida and Nelson 2009, Roberts et al. 2017), transportation infrastructure management and rural access (Iimi et al. 2016, Worldbank 2016), resource allocation strategies (Islam et al. 2006, Deichmann et al. 2011), disaster risk mitigation, management and reduction (Ehrlich et al. 2018a, Aubrecht et al. 2016, Gunasekera et al. 2015, Mondal and Tatem 2012, Taramelli et al., 2010), climate change research (Blankespoor, Dasgupta and Lange 2017, Dasgupta et al. 2011, McGranahan et al. 2007), sampling design for household surveys (Blankespoor et al. 2018, Thomson et al. 2017), public health campaigns and assessments (Snow et al. 1999, Hay et al. 2004, Jones et al. 2008, Weber et al. 2018, Dunn et al. 2019) and sustainable resource management (Koch et al. 2008, Parish et al. 2012, McDonald et al. 2011) among many other applications[1]. International frameworks for development and sustainability depend on the availability of population data, which are commonly used as a denominator in calculating different metrics and indicators. Such frameworks include the Sustainable Development Goals (SDGs), the Sendai Framework for Disaster Risk Reduction, the UNFCCC Paris Agreement, and the United Nations New Urban Agenda, to mention just a few.

The field has seen advances at multiple levels. First, the spatial resolution of underlying census data available for geoprocessing, along with the standards for producing such data (United Nations 2009), has improved dramatically in many countries since the creation of the earliest gridded population data products such as the Gridded Population of the World version 1 (Tobler et al. 1995, Deichmann 1996). Second, significant progress has been made through advances in information extraction and classification of populated land area from remote sensing data at various resolutions (Wardrop et al. 2018). The increased availability and spatial granularity of remotely sensed information about topography, vegetation and land cover has been critical to improve the identification of such places that are potentially inhabited and even the estimation of counts of people living there (Frye et al. 2018, Nieves et al. 2017, Pesaresi et al. 2013). Third, the combination of access, increased

---

[1] The list of citations here are just a few of hundreds of applications that could have been identified. This paper does not aim to be a complete review of the literature or applications of all usages of the gridded data products under review. Links to citations of particular data products are found in Section 6 below.

computing power, and greater spatial accuracy in ancillary datasets has provided the basis for methodological advances to redistribute census-enumerated population counts to grid cells at continental and global scales with high accuracy (e.g., Freire et al. 2018) and to create time series of population estimates that can be used to fill in data gaps between national census surveys that are commonly taken at decadal intervals (e.g., WorldPop and CIESIN 2018).

As a result of these recent developments, there are now several global and continental gridded population data sets that are based on different modeling approaches and input data layers. As might be expected, there are similarities but also important differences among these products, and yet to date there has neither been a systematic review of these various approaches, nor a comparison of the corresponding outputs. This represents a serious gap in the literature as these differences can easily lead to misunderstandings or inappropriate use of population grids. The objective of the paper is to fill that gap by helping guide

users in forms of appropriate, uncertainty-aware use of the available global gridded population datasets in different application areas. Such an assessment is necessary as knowledge of underlying approaches and input data can inform about what each gridded product actually measures. For example, the exposure of a target population to disasters requires a population grid that 1) covers the area of interest, 2) provides a meaningful analytical unit (i.e., the size of the grid cell), 3) warrants the temporal currency needed relative to the time of interest, and 4) estimates the correct target population.[2] This example demonstrates

why applying population grids is not trivial; grids have different characteristics that may affect the accuracy and precision of the analysis but also their suitability in a given context.

The above aspects together provide the essential components to assess the fitness for use of a data product in the context of relative data quality (Tayi and Ballou 1998). Fitness for use is a concept that has often been used to assess the appropriateness of a given spatial dataset for an intended purpose (Agumya and Hunter 1999, de Bruin, Bregt and Ven 2001, Devillers et al.

2007). Here, this concept will be applied to guide a growing user community in making informed decisions regarding the most appropriate dataset(s) for their intended use by better understanding the characteristics of the available different data products that also include the modeling assumptions behind them. Spatial, thematic and temporal accuracy play a key role in formalizing fitness for use. However, the multidimensionality of accuracy in the case of population grids is further driven by the nature and heterogeneity of the input population data, the use and characteristics of ancillary data involved and the methodological

framework applied to redistribute population counts to grid cells. All these factors will be systematically explored in this article.

This review targets researchers and applied users in the geospatial, demographic, environmental and land use research communities with diverse needs. Section 2 begins with a brief history of population gridding. Section 3 looks at commonalities and differences in methods applied and ancillary data used to produce gridded population data. Section 4 provides an

introduction to the data products of interest herein, and summarizes the approaches behind the most recently released global as well as some selected regional and national gridded population datasets. Section 5 provides a comparative discussion of several components related to the fitness for use of the different data products. Finally, we list guidelines that can help the user community make informed decisions related to the fitness of a given population data product for their intended use and identify future avenues of work and needed investments in Section 6.

**2 People as gridded distribution: Background and historical development**

In the past, mapping population typically entailed linking tabulated population statistics to "vector features", such as points (for example, geographic coordinates indicating city centers) and/or polygons (most notably, administrative units or census

---

[2] In the production of gridded population data, the underlying census data are accepted as demographically accurate. While demographers concern themselves with such issues as age-heaping (Myers, 1993) or completeness of registrations or census-samples (e.g., Potter and Ordóñez, 1976) at the national and first-order administrative level, to the extent that such problems exist (perhaps to an even greater degree) in the fine-grain, underlying spatially-refined data, these issues are inherited into the gridded data products.

enumeration areas). Beginning in the 1990s, a new approach to mapping population distributions emerged, which was to convert population data from irregular vector formats to gridded surfaces composed of regular, standardized grid cells or pixels (e.g., Martin and Bracken 1991, Tobler et al. 1995, Martin 1996, Balk et al. 2006, Thomson et al. 2017).

The impetus to grid population data arose soon after the first GIS software packages were developed, and as the spatially-oriented research community began to use a growing number of gridded biophysical and geophysical data products. Regular grids represented an efficient and consistent data storage format, and the move to gridded data - already in use by the climatological modeling community - was reinforced by the growing array of remote sensing data products that began to appear in the 1970s and 1980s. By gridding population, researchers were able to more easily integrate population count and density data with biophysical data to better understand spatial distributions and components of socio-environmental systems. Furthermore, by decoupling the data from their original administrative boundaries, populations could then be easily aggregated to different units of interest (e.g., watersheds or climate zones) for spatial and statistical analysis (Balk et al. 2009).

Early efforts to grid populations include an African population grid for UNEP's Global Atlas of Desertification (Deichmann and Eklundh 1991), the NASA Goddard Institute for Space Studies' Global Distribution of 1984 Population Density at 1°×1 ° Resolution (Fung et al. 1991), and Tobler's pycnophylactic method (Tobler et al. 1997), which resulted in the first version of Gridded Population of the World in 1995. These early approaches spread populations evenly across grid cells within input census units, with adjustment effects applied (in the case of the pycnophylactic method) at the unit boundaries. One inherent problem of these early modeled outputs is the existence of aggregation effects that often lead to analytical challenges, as described in the next paragraph. Two concomitant changes helped to partially overcome this inherent problem: First, improvement in the spatial resolution of the underlying population data, and increased computation capacity to use higher-resolution data, have reduced the impact of this problem for many applications. Second, as methods and data availability have progressed, researchers also sought to improve the spatial resolution of population estimates by reallocating populations using ancillary datasets, a spatial refinement strategy known as dasymetric mapping (Semenov-Tian-Shansky 1928, Wright 1936), in combination with different statistical methods (e.g., Wu et al. 2005). Both dasymetric and statistical techniques continue to play an important role in gridded population mapping (Mennis 2009), as discussed below. In addition to such spatial refinement strategies, ongoing efforts also focus on improving the temporal coverage and temporal resolution as well as increasing the variety of population characteristics mapped.

While the development of consistent, comparable grids is what makes gridded data products so useful, there are some important implications that need to be addressed, as should be the case for any geospatial data. Population is not randomly distributed and therefore the allocation and representation of populations will always be subject to aggregation effects. These effects have been described in the geography literature as the Modifiable Areal Unit Problem (MAUP) (Openshaw and Taylor 1981). According to MAUP, the level of aggregation -- in this case the census unit or administrative level -- and the shape of the reporting units can affect the analysis in ways that are difficult to predict. MAUP is manifested in the flawed assumption of homogeneity of population distributions across census reporting units. The spatial resolution of a gridded population dataset determines the output analytical unit and thus will have implications due to these same aggregation effects after transitioning population counts from vector boundaries to grid cells. In other words, these aggregation-related problems of enumerating data are not eliminated but are propagated into a different data structure through the creation of gridded population data.

As one of the most persistent problems in geographical analysis, MAUP-related research has made significant progress to better understand the sensitivity of analytical results due to changing aggregation levels using synthetic and real-world data (Amrhein 1995, Steel and Holt 1996, Flowerdew et al. 2001, Pawitan and Steel 2006, Wong 2009, Arbia and Petrarca 2011, Maclaurin et al. 2015). However, because of this sensitivity, it's important to recognize that MAUP affects the fitness for use of data products for specific analyses in which the spatial precision of population locations is critical. Other implications that affect the quality of population grids have been reported by the data producers including temporal differences of input and ancillary variables as well as the measurement construct of population that is mapped. While these quality aspects are important

to help the user community by guiding general applications, the impact of these aspects on the fitness for use of the data products for specific applications is difficult to measure and not well understood.

## 3. Putting people in places: Key methods and ancillary data

### 3.1 Methods for population redistribution

Understanding the fundamentals of the different data integration approaches is an important aspect in evaluating the fitness of any given dataset for specific uses or cases. The process of gridded population mapping requires reallocation of spatial data from "source" units into "target" units, usually as a form of disaggregation that can be done through different approaches including various forms of areal interpolation and statistical modeling.

**Areal weighting** techniques (the simplest form of areal interpolation, also known as proportional reallocation) evenly re-
distribute source data into target grid cells based on proportions of overlap with no ancillary data input informing the process (Goodchild and Lam 1980, Mennis and Hultgren 2006) (Figure 1a). The source input data may be census-based or other administrative data and the target grid cell represents a spatial unit which is generally smaller than the source units. An assumption associated with this approach is that the population is uniformly redistributed from the source units to target cells that overlap with the source units. This assumption is a gross simplification as population distributions are not uniform, but
the approach is computationally efficient and simple in creating spatially-explicit and globally consistent population estimates. Such products are well suited for informing policy-making efforts that do not require fine spatial resolution (Doxsey-Whitfield et al. 2015), or for performing correlation analyses in which endogeneity issues are excluded (e.g., Cohen & Small 1998). The Gridded Population of the World (GPWv4) product is an example of this approach.

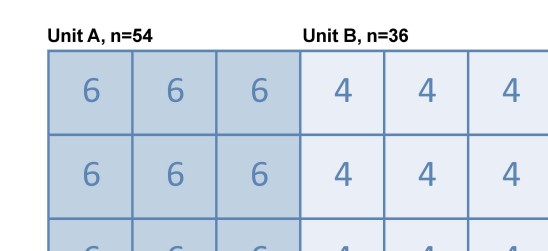

**Figure 1. Schematic illustration of different types of techniques for population redistribution or allocation from source to target grid cells: (a) Areal weighting as the simplest form of areal interpolation that does not use any ancillary variables; (b) Dasymetric mapping using binary ancillary variables that inform and refine areal weighting; (c) Dasymetric mapping using varying population weights that may be empirically derived or based on set rules; (d) Statistical modelling to estimate relationships that can be used for population modelling. The different grey tones in (b)-(d) indicate different underlying data informing the areal interpolation process.**

When ancillary data informs the redistribution through areal interpolation from source area to target cell, the technique is referred to as dasymetric mapping (Semenov-Tian-Shansky 1928, Wright 1936, Eicher and Brewer 2001, Mennis 2003, Mennis and Hultgren 2006). The ancillary variables, often produced and available at finer spatial detail than the input population data, can be used to develop weighting schemes for reallocating population from the source area to target units

depending on existing or assumed relationships between the two. Ancillary variables can include land cover, topography, land-use zones, street networks, remote sensing data and more (for details and more examples see Zandbergen and Ignizio 2010, Nieves et al. 2017; for an overview see Mennis 2009). For example, redistributing population from a source area (e.g., a census tract) that includes built or developed parts along with forest and agricultural land uses will more heavily weight the built area in redistributing population counts because it is more likely that these areas are populated (Mennis and Hultgren 2006, Bhaduri

et al. 2014). All dasymetric mapping approaches rely on existing relationships between population (e.g., provided by the input census data) and ancillary information (e.g., land cover) that can be exploited to redistribute population to finer spatial units with higher accuracy. More **traditional dasymetric approaches** vary in the allocation method applied, ranging from binary dasymetric refinement (Figure 1b, Eicher and Brewer 2001) to more complex weighting approaches (Figure 1c) such as 'intelligent' dasymetric mapping (Mennis and Hultgren 2006). These approaches differ in the way relationships between

population and ancillary variables are derived (i.e., presence/absence based, empirically derived or optimized) to determine weights for different locations to inform the disaggregation of the population totals.

Several **statistical modeling** approaches have been described in the literature that blur the line between statistical analysis and dasymetric mapping, and can be viewed as another means of population estimation, traditionally focusing on the problem of small area estimation (e.g., Birkin and Clarke 1988, Wong 1992, Bogaert 2002) or as a type of dasymetric refinement (e.g.,

Mrozinski and Cromley 1999, Leyk et al. 2013) (Figure 1d). The difference to more traditional approaches is that the weights are statistically derived by regressing population counts or densities against various types of predictive variables (Mennis 2009), derived from ancillary data layers such as density or length of streets (Reibel and Bufalino 2005), or remotely-sensed data (Harvey 2002, Wu et al. 2005).

More recently, an increasing number of **hybrid approaches** have been described that explicitly combine the more traditional

concept of dasymetric mapping with statistical analytical frameworks. These approaches often rely on machine learning techniques or ensemble prediction that enable the robust estimation of population weights and, in a subsequent step, inform a dasymetric redistribution process (Nagle et al. 2014; Stevens et al. 2015). For example, a statistical model (e.g., a Maximum-Entropy approach or a Random Forest model) estimates a population density layer. These estimated population densities provide a weighting layer that is then used to dasymetrically redistribute total population counts within each source unit to its

target grid cells. If there exists a robust settlement layer, then the hybrid approach would use the statistical weighting layer to dasymetrically redistribute the total source zone population counts only to target grid cells that are classified as settlements (Reed et al., 2018). Such hybrid dasymetric approaches have shown promising results when compared to other techniques for producing gridded population maps (Sorichetta et al. 2015, Reed et al. 2018).

**3.2 Ancillary data**

The products included in this comparative review are the outcomes of different data integration approaches to produce gridded population distribution datasets based on different techniques of refinement, zonal statistics, reallocation or inter- and extrapolation. Different ancillary data have been used in slightly different ways to create different population models. As mentioned, all ancillary data have in common that they exhibit some kind of relationship to population that can be exploited in population redistribution models to increase the accuracy of population estimates. These relationships may be of correlative

nature, based on empiric rules or even binary. While the literature on population modeling and dasymetric mapping has described a variety of such ancillary variables, the data that can be used in national, regional and global population grid production has to be available consistently for large extents, and for different points in time, thus limiting the choices for

researchers and data producers. One important class of ancillary data is that of urban land use area or human settlements detections. Figure 2 provides an overview of this type of ancillary data available at different points in time including satellite images (Landsat, MODIS), land cover products and settlement layers (e.g., GHSL or the Global Urban Footprint (GUF+)) in relation to commonly available census data. This overview highlights apparent temporal offsets between input population data and some ancillary data. It also emphasizes the high temporal resolution of satellite data, which can have varying quality due to cloud cover and other characteristics and provides the basis for the derivation of abundant ancillary variables including land cover and settlement data.

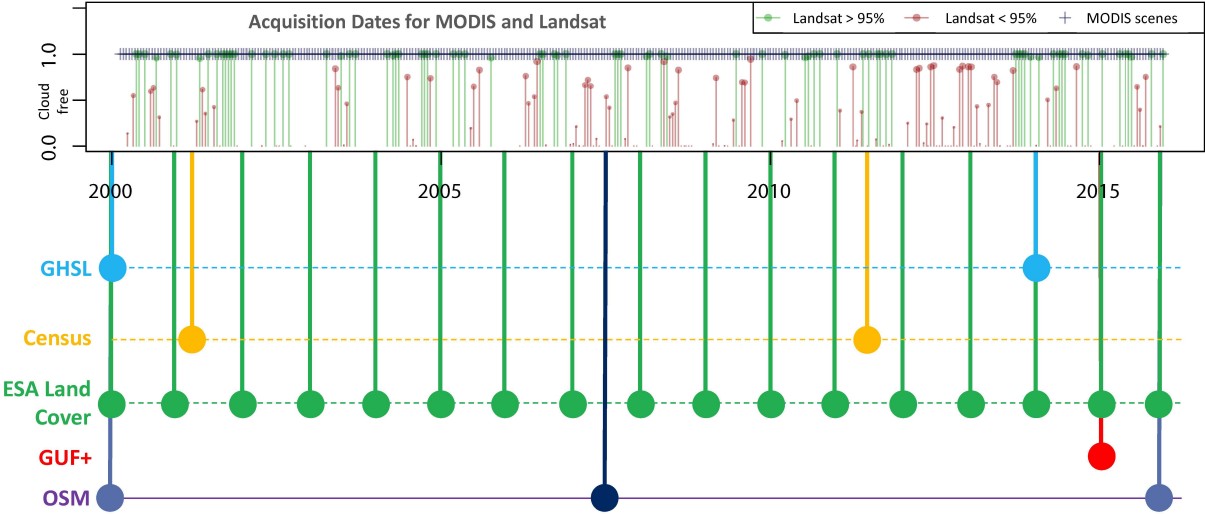

**Figure 2. Identification of different ancillary data that inform spatial and temporal interpolation approaches to create gridded population data across scales of interest. Temporal fidelity in the Landsat (30m resolution; with varying proportions of cloud-free area) and MODIS (250m resolution) sensors are shown in relation to typical points in time for censuses alongside several derived ancillary data products such as the European Space Agency (ESA) annual land cover data (300m resolution), and the Global Human Settlement Layer (38m resolution) at various publication dates. The Global Urban Footprint (GUF+) exists for one point in time only. Also noted are OpenStreetMap data, vector-based information that is increasingly explored as a possible ancillary data source, which can be acquired anytime and is potentially useful for more contemporary time periods as a static variable; as it is continually evolving, it's currency may deviate by region.**

Table 1 summarizes the input variables, including these land-use type and other ancillary data, used to create the different products (also available at https://www.popgrid.org/compare-data); as described earlier, Table 2 provides additional information on the modeling methods used.

**Table 1. Summary of input variables used in modeling gridded population, globally.**

| Gridded Population Dataset | Population | Ancillary Data Layers | | | | | | | | |
| | | Roads | Land Cover | Built structures | Cities or Urban areas | Night-time lights | Infrastructure | Environmental data[b] | Protected areas[a] | Water bodies |
| --- | --- | --- | --- | --- | --- | --- | --- | --- | --- | --- |
| GPW | X | | | | | | | | a | X |
| GRUMP | X | | | | X | X | | | a | X |
| LandScan | X | X | X | X | X | | X | X | X | X |
| GHS-POP | X | | | X | | | | | | |
| WPE | X | X | X | | X | | | | | X |
| WorldPop | X | X | X | X | X | X | X | X | X | X |
| HYDE 1950- | X | | | | | | | X | | X |

*[a] Protected areas were not masked out, but national statistical offices often assign no data or 0 (zero) to protected areas;*

*[b] Climate, topography, elevation*

### 3.3 Different methods and sources of uncertainty

Figure 3 illustrates, using a region in Kenya, how different ancillary data layers, typically used for population redistribution including roads, land cover, protected areas and topography (Figure 3b-e) affect the resulting population distribution (Figure 3f). Different methods described above will employ these variables in different ways and operate under varying assumptions, and often result in different estimates. Thus, there are expected relationships and trends that can be observed for most population grids. For example, low road density, rough topography and high elevations, the presence of protected area and non-urban land cover are commonly related to low population densities. However, Figure 4 illustrates remarkable differences between the population distributions of the data products described in this review for a larger area in Kenya, highlighting the importance of informing the user about critical aspects and characteristics of the different data layers. Note that in Figure 4a population counts (not density) are rendered per irregularly shaped level-5 census unit. In Figure 4c-h, population is rendered per grid cell. Note that the grid cell size is not the same across the panels and is specific to each data product. Within each panel, however, the grid cells have the same extent and can be interpreted as population densities.

It is important to acknowledge error accompanying the estimation results from such redistribution approaches. This includes uncertainty associated with the original census, the areal aggregation of both the input census data and the ancillary data products (Wu et al. 2005), and the model used to estimate statistical relationships (Nagle et al. 2014, Sinha et al. 2019). Recent research has increasingly stressed the complexity of uncertainty in such applications as well as the difficulty to carry out validation due to the lack of reference data (Mennis and Hultgren 2006, Zandbergen and Ignizio 2010). Therefore, error assessments tend to appear mostly in studies in data-rich settings.

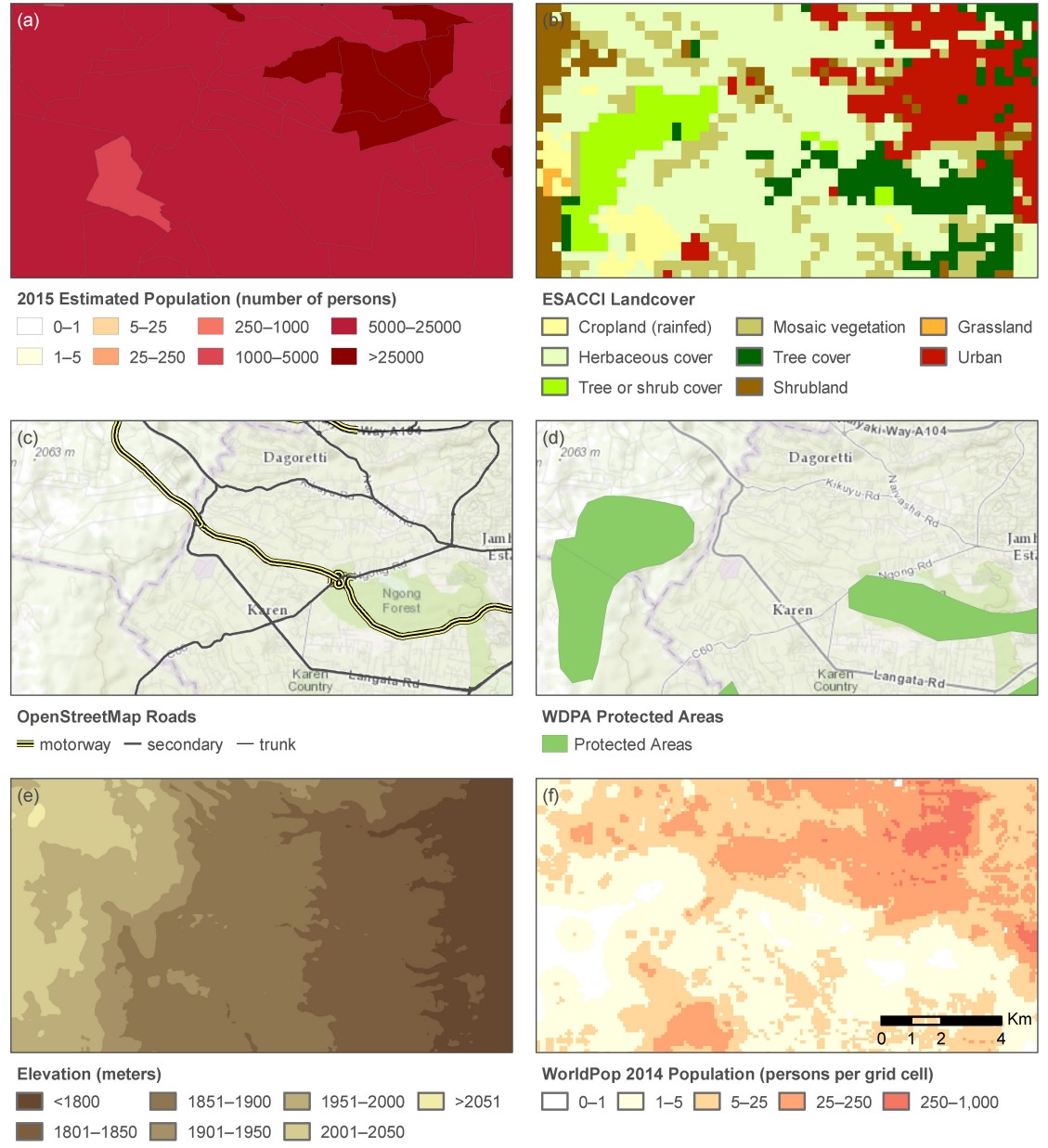

**Figure 3. A schematic illustration of refinement effects of ancillary data layers on census population data to create gridded population grids at fine spatial resolution for a small study area near Nairobi, Kenya: (a) Kenya National Bureau of Statistics, Population and Housing Census 2009, level 5 population units (Center for Development and Environment, Kenyan Atlas Project) as input, (b) European Space Agency (ESA) Climate Change Initiative (CCI) Land Cover 2015 (300 m resolution), (c) OpenStreetMap major roads, (d) World Database on Protected Areas (March 2019 Release), (e) Viewfinder Panoramas 3 Arc seconds Digital Elevation Model, (f) WorldPop 2014 Population Count (100 m resolution) as one exemplary population grid created.**

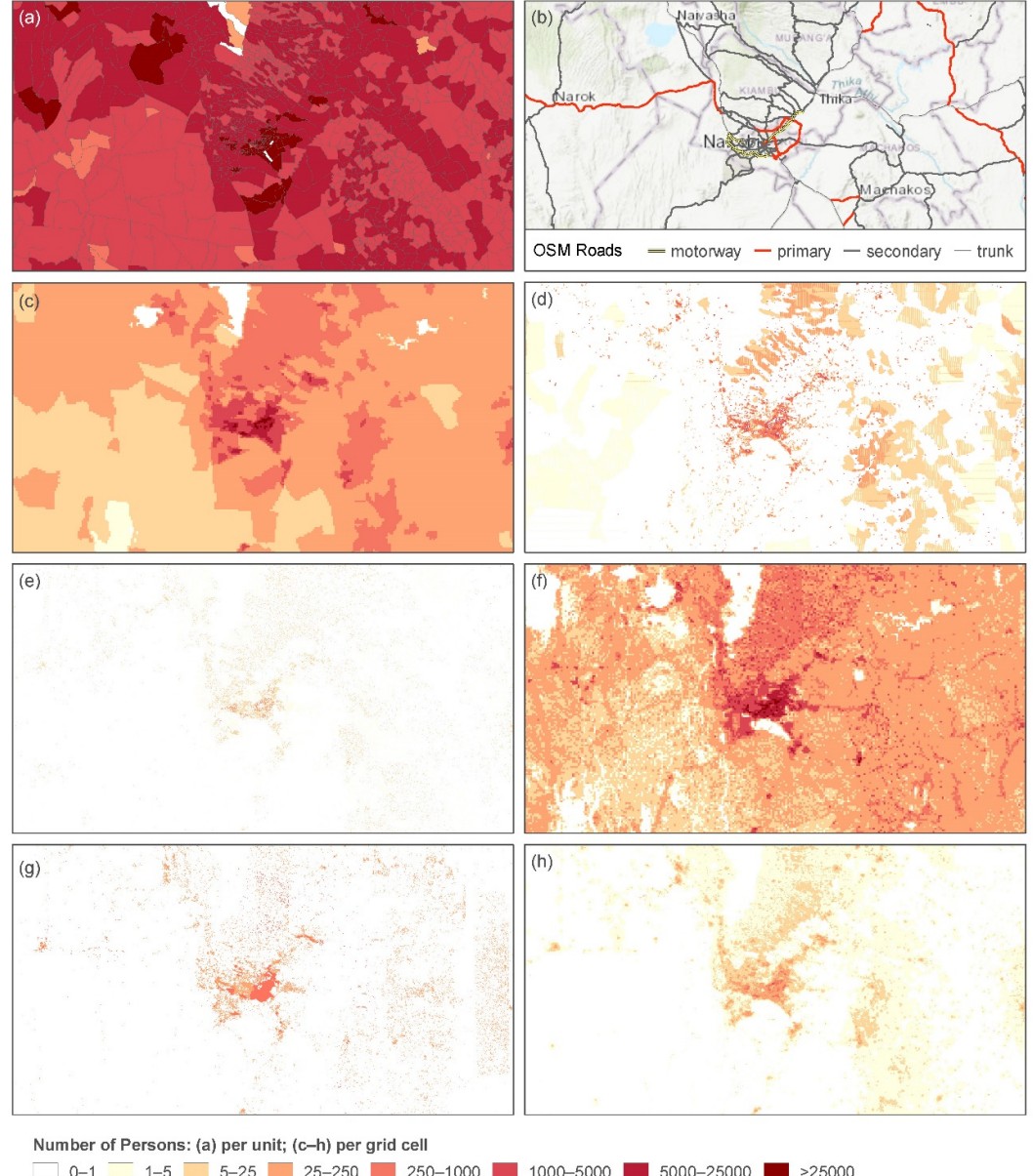

**Number of Persons: (a) per unit; (c–h) per grid cell**

0–1  1–5  5–25  25–250  250–1000  1000–5000  5000–25000  >25000

**Figure 4: Illustration of population input, exemplary ancillary data and different outcome data for a larger region around Nairobi, Kenya: (a) Kenya National Bureau of Statistics, Population and Housing Census 2009, level 5 population units (Center for Development and Environment, Kenyan Atlas Project), (b) basemap with roads and topography as ancillary data, (c) Gridded**
5  **Population of the World version 4 Revision 10, UN Adjusted 2015 Population Count (1 km), (d) Global Human Settlement Layer 2015 Population Count (250 m), (e) High Resolution Settlement Layer 2015 Population Count (30 m), (f) Landscan 2015 Population Count (1 km), (g) Esri World Population Estimate 2016 Population Count (150 m), (h) WorldPop 2014 Population Count (100 m).**

The persistent challenges with modeling and validating gridded population datasets especially in data-poor regions has driven more recent initiatives that focus on modeling gridded population from the ground up, relying on micro-census data and
10  geostatistical covariates in a statistical modeling framework (Wardrop et al. 2018). Such techniques, in the absence of reliable or recent census data, leverage advances in computational and statistical frameworks along with increased spatial fidelity of remotely sensed products and advances in global positioning system (GPS)-enabled field survey techniques to produce gridded population surfaces. This type of approach is considered complementary to more traditional, census enumeration-based efforts.

**4 Current data products, characteristics and availability**

This section summarizes several global data products including the Center for International Earth Science Information Network (CIESIN)'s Gridded Population of the World (GPWv4.11) and Global Rural Urban Mapping Project (GRUMPv1); The European Commission Joint Research Centre (JRC) and CIESIN's Global Human Settlement Population Layer (GHS-POP);

5   Oak Ridge National Laboratory's LandScan; ESRI's World Population Estimate (WPE); and WorldPop's WorldPop datasets. We also reference The History Database of the Global Environment (HYDE) as a gridded data product representing a long-term historical context (i.e. ~12,000 years). Depending on the estimation method applied and ancillary data used, these different data products can be seen as unmodeled, slightly modeled and highly modeled population grids. While the focus of this review is on global population grids, we also discuss a number of country and regional/continental grids, including Facebook and

10   CIESIN's High Resolution Settlement Layer (HRSL), JRC's European GHS Population Grid, and the U.S. Census Bureau's country grids (Demobase). Owing to space constraints, we omit gridded population projections such as those developed by Jones and O'Neill (2016). Similarities and differences in these data products are detailed in Table 2. Extended data documentation and visual comparison tools (tables and map services) are available through the POPGRID website (www.popgrid.org).

**Table 2. Detailed characteristics and availability of the datasets described in this review. More information about these and other data products can be found at https://www.popgrid.org/popgrid_files/popgrid-data-comparison-tables_0.pdf.**

| | Dataset | Source | Population concept | Method | Spatial Resolution | Year(s) Represented | National level population totals | Distribution Policy | Reference Link |
|---|---|---|---|---|---|---|---|---|---|
| | | | | | | GLOBAL POPULATION GRIDS | | | |
| Unmodeled | Gridded Population of the World (GPWv4.11) | Center for International Earth Science Information Network (CIESIN), Columbia University | De jure / de facto | GPW1: pycnophylactic; GPW2,3,4: areal weighting | 1 km (v4) | 2000; 2005; 2010; 2015; 2020 | 2 versions: 1) official country census totals; 2) country totals adjusted to United Nations Population Division (UNPD) estimates and projections | Open access | http://sedac.ciesin.columbia.edu/data/collection/gpw-v4 |
| Lightly modeled | Global Rural Urban Mapping Project (GRUMPv1) | CIESIN, Columbia University; International Food Policy Research Institute, The World Bank, Centro Internacional de Agricultural Tropical | De jure / de facto | Dasymetric | 1 km | 1990; 1995; 2000 | UNPD estimates and projections | Open access | http://sedac.ciesin.columbia.edu/data/collection/grump-v1 |
| | Global Human Settlement Layer - Population (GHS-POP) | European Commission Joint Research Centre (JRC) and CIESIN, Columbia University | De jure / de facto | Dasymetric refinement, proportional to built-up density | 250 m | 1975; 1990; 2000; 2015 | UNPD estimates and projections | Open access | http://ghsl.jrc.ec.europa.eu/ghs_pop.php |
| Highly modeled | LandScan Global Population Database (Landscan Global) | Oak Ridge National Laboratory (ORNL) | Ambient (day-time) | Smart interpolation | 30 arc-seconds | annual releases 2000–2016 | US Census Bureau | Open for research; commercial use at cost | https://landscan.ornl.gov/ |
| | WorldPop | WorldPop, University of Southampton | De jure / de facto | Statistical / dasymetric | 100 m | 2000-2020 | 2 versions: 1) Country-official estimates, and 2) UNPD estimates and projections | Open access | www.worldpop.org |
| | World Population Estimate (WPE) | Environmental Systems Research Institute (ESRI) | Combined (de jure, defacto, estimates) | Dasymetric Redistribution (Smart) | 250 m 250 m 150 m | 2013, 2015, and 2016 | Country-official estimates with 134 countries processed further by M. Bauer Research GmbH | Free to ArcGIS Users | https://sites.google.com/ciesin.columbia.edu/popgrid/find-data/esri |
| | History Database of the Global Environment (HYDE) Population Grids v3.2 | Netherlands Environmental Assessment Agency (PBL) | De jure / de facto population | Dasymetric mapping using historical population, cropland, pasture data, satellite data | 5 arc-min (ca. 10km) | 10,000 BC - 2015 | United Nations World Population Prospects (2008 Revision) after 1950 | Open access | https://themasites.pbl.nl/tridion/en/themasites/hyde/download/index-2.html |
| | | | | | REGIONAL/ CONTINENTAL POPULATION GRIDS | | | |

| | | | | | | | | |
|---|---|---|---|---|---|---|---|---|
| **High Resolution Settlement Layer (HRSL)** | Facebook Connectivity Lab and CIESIN | De jure / de facto population | Binary Dasymetric | 30 m (1 arc-second) | 2015 | Country-official estimates of more than 30 countries | Open access | https://ciesin.columbia.edu/data/hrsl/ |
| **European GHS Population Grid (GHS-POP-EUROSTAT)** | European Commission Joint Research Centre (JRC) | De jure / de facto population | Intelligent dasymetric mapping | 100 m | 2011 | Country-official estimates | Open access | http://data.jrc.ec.europa.eu/dataset/jrc-ghsl-ghs_pop_eurostat_europe_r2016a |
| **Gridded Population Mapping (Demobase)** | U.S. Census Bureau | de jure population | Statistical / dasymetric | 100 m | Depends on country (1998-present) | U.S. Census Bureau International Data Base and national censuses | Open access | https://www.census.gov/geographies/mapping-files/time-series/demo/international-programs/demobase.html |

## 4.1 Global population data production efforts

**Gridded Population of the World version 4 (GPW4)** is a data collection consisting of gridded data products on total population counts and densities and other key demographic variables, globally at a nominal spatial resolution of 1km using the World Geodetic System (WGS84) as geographic reference system (Doxsey-Whitfield et al. 2015). GPW4 includes estimates for the years 2000, 2005, 2010, 2015, and 2020 respectively. Additionally, GPW4 includes vector point data representing the centroids of input census enumeration units, and gridded data on land and water area estimates, national identifiers, and data quality metrics. GPW4 employs a uniform allocation approach to disaggregate population which is based purely on the land area of a given pixel (unmodeled, see Table 2). The Mean Input Administrative Area can be used as a data quality metric to provide users with guidance as to the effective local resolution of original input population data. Because the size and extent of input census geographies is highly variable, within and across countries, the scale at which GPW4 data should be analyzed differs by region. For example, in the USA, where Census blocks are the primary input units, highly localized analysis is appropriate, whereas the coarse input geographies of Libya require aggregations to provincial scales for analysis. Two variants of the population grids are available: those based solely on inputs from the data supplier (typically national statistical offices), and national totals that match the total population estimate of the United Nations' World Population Prospects (2019). Detailed documentation and metadata on nominal resolution and sources of input data are provided. These data are freely accessible and downloadable at: http://sedac.ciesin.columbia.edu/data/collection/gpw-v4.

The **Global Rural Urban Mapping Project, v1 (GRUMP)** data collection builds on GPW, also in WGS84 and at a nominal resolution of 1km, with the explicit aim to capture urban locations and populations and to distinguish those from surrounding rural areas. The collection consists of global data sets normalized to the years 2000, 1995, and 1990 that indicate urban settlement points and grids of urban extents, as well as population count and density grids that are lightly modeled, taking the urban location information into account (Balk et al. 2005, Balk 2009). Using the stable-city lights data from the National Oceanic and Atmospheric Administration (Elvidge et al. 1997), GRUMP was the first global database to render urban areas spatially and connect those locations with estimates of population. Although newer night time light time-series data are now available (e.g., Elvidge et al. 2017), for a variety of reasons, updates to this exact data product are not presently expected. This is partly due to the fact that the time-series does not extend as far back as other possible settlement input layers, and that more recent night-lights can be better put to use as an independent proxy for economic activity rather than urban location. The data collection is freely available at https://sedac.ciesin.columbia.edu/data/collection/grump-v1.

The **Global Human Settlement Population Grid (GHS-POP)** depicts the distribution and density of the total population as the number of people per grid cell (250m spatial resolution) in World Mollweide equal-area projection. Residential population estimates (counts) per smallest census units available, used also by CIESIN GPWv4 for the years of interest, are disaggregated to grid cells, directly (linearly) proportional to the ratio of built-up areas within a cell to the total cell surface (Freire et al. 2016, 2018). Global mapping of built-up areas was performed through the Global Human Settlement Layer (GHSL) project using Landsat imagery collections for nominal epochs 1975, 1990, 2000 and 2014 (Pesaresi et al. 2013, 2016a, 2016b). The

GHSL approach is grounded on the concept that buildings and their agglomerations (i.e., settlements) are nowadays the main visible and direct manifestation of human presence (and activity) on the Earth's surface. GHS-POP aims to constitute a detailed and consistent time series of lightly modeled population distributions that is based on reproducible methods for sustainable data production (Melchiorri et al. 2019) and can be used in policy support in numerous domains (Ehrlich et al. 2018b). These grids are created using open and free input data and are also freely accessible and downloadable at: https://ghslsys.jrc.ec.europa.eu/ghs_pop.php.

Oak Ridge National Laboratory's **LandScan Global** represents an ambient (average day/night) population distribution in a 30 arc-second (~1 km) resolution grid using the World Geodetic System (WGS84) for spatial reference (Dobson et al. 2000). LandScan uses census and other geographic data, as well as remote sensing imagery in a multivariate dasymetric modeling framework to disaggregate census counts within administrative boundaries (Dobson et al. 2003, Bhaduri et al. 2002). The final product displays a combination of locally adaptive models tailored to match input geographies and different environmental conditions in countries and regions. The modeling approach, defined as "smart interpolation," uses subnational level census counts for each country and ancillary datasets, including land cover, roads, slope, urban areas, village locations, and high resolution image classifications; all of which are key indicators of population distributions. Based upon the spatial data and the socioeconomic and cultural understanding of an area, cells are preferentially weighted for the possible occurrence of population during the course of a day. Within each country, the population distribution model calculates a "likelihood" coefficient for each cell and applies the coefficients to the census counts, which are employed as control totals for respective areas. The total population for that area is then allocated to each cell proportionally to the calculated population coefficient to compute counts of ambient or average day/night population. LandScan Global is available for download free of charge to the educational community at https://landscan.ornl.gov/.

Esri's **World Population Estimate (WPE)**, initiated in 2014 and produced at the Environmental Systems Research Institute (ESRI), includes population count and density grids at a spatial resolution of 150 meters, referenced through the WGS84 geographic coordinate system (Frye et al. 2018). WPE is based on the dasymetric re-distribution of human population data enumerated within the most detailed census data available for each country to raster cells using a raster model representing the footprint of human settlement (Frye et al. 2018). The footprint of human settlement is produced using various ancillary data layers. First, base scores are derived through the combination of a 30-meter resolution global classified land cover dataset (MacDonald Dettwiler and Associates (MDA) 2017), road intersection points (HERE 2019, OpenStreetMap Foundation (OSMF) 2015), and populated place points from GeoNames (GeoNames 2013). The base scores are augmented with texture scores derived from 15-meter resolution Landsat 8 panchromatic images using a rugosity (i.e., terrain roughness) model (Jenness 2004). The base scores are used to allocate population to WPE cells to create gridded representations of estimates of population counts, population density (number of persons per square kilometer), the likelihood of settlement, as well as confidence scores. WPE is the only commercial product described, available through https://www.arcgis.com/home/item.html?id=92d3005feb84428a8f85160f2451ec63.

The **WorldPop** program produces a variety of demographic gridded data products at the global and country scales (Tatem 2017), including population counts, within 3 arc-seconds grid cells (~100m at the equator) in the Geographic projection WGS84 (Stevens et al. 2015). Initiated in October 2013, the WorldPop project replaces and merges the regional AfriPop (Linard and Tatem 2012), AsiaPop (Gaughan et al. 2013) and AmeriPop (Sorichetta et al. 2015) population mapping projects. The main method for producing WorldPop products is a weighted dasymetric approach that relies on a random forest model (Breiman, 2001) to produce a predictive weighting layer for dasymetrically redistributing population counts into gridded cells (Stevens et al. 2015). Individual country outputs from the WorldPop project provide an open access, transparently documented archive of spatial demographic datasets for many regions in the world including Central and South America, Africa and Asia

to support development, disaster response and health applications (Gaughan et al. 2013, Stevens et al. 2015, Sorichetta et al. 2015, 2016). In addition, the WorldPop program produces a standardized, temporally and spatially consistent set of gridded products at the global scale. These freely available datasets include the input population data and covariates used in model prediction (Lloyd et al. 2017), annual gridded population count datasets also structured by 36 age/sex classes from 2000 to 2020, and grid cell area estimates that can be used to derive gridded population density datasets. All data can be downloaded from www.worldpop.org.

The **History Database of the Global Environment (HYDE)** includes maps of historical estimates of total, urban and rural population, population density and built-up area at a spatial resolution of 5 min longitude/latitude, provided in decimal degrees. HYDE covers a time period from 10,000 before Common Era (BCE) to 2015 Common Era (CE) and is described as an internally consistent combination of historical population estimates and allocation algorithms with time-dependent weight maps for land use (Klein Goldewijk et al. 2010, 2011 and 2017). For the period prior to 1950, historical input population estimates were taken from the general literature and supplemented with the sub-national population numbers and country-specific sources to build time series for each province or state of every country. For the period after 1950, the underlying input data is based on 1950-2015 population estimates from the United Nations World Population Prospects (2008 Revision) as well as land cover and land use data products. All data can be downloaded from https://doi.org/10.17026/dans-25g-gez3.

### 4.2 National and regional/continental population data production efforts

It is imperative for a review of existing global population data products to also reference production efforts at national, regional or continental scales that often make use of more detailed input data but are based on similar methodological frameworks. Such country- and regional-level products are often created for specific purposes, which may influence the decision rules applied for their creation. Often these data products are based on more up-to-date ancillary and input population data and thus may provide pointers for future global population data creation once those ancillary data could become available worldwide.

For example, Facebook Connectivity Lab and CIESIN's **High Resolution Settlement Layer (HRSL)** provides estimates of human population distribution in 33 countries in Central- and South America, Africa and South-East Asia, at a resolution of 1 arc-second (approximately 30 m), in the Geographic projection WGS84 for the year 2015. Machine learning techniques are used to identify potentially populated areas (settlement) using very high resolution satellite imagery. Proportional allocation is then applied to redistribute population from recent census data onto grid cells identified as settlement extent (Tiecke 2016, Tiecke et al. 2017). This data production effort was driven mostly by Facebook's interest in locating people in remote areas of developing countries such as Burkina Faso, Ghana, Haiti and Sri Lanka who may be in need of internet access and is available from: https://www.ciesin.columbia.edu/data/hrsl/.

Developed by the European Commission for the purpose of producing the most detailed possible population grid for policy analysis and support, the **European Global Human Settlement (GHS) population grid** represents the distribution and density of total residential population, expressed as the number of people per grid cell (100 m spatial resolution) in equal-area projection (LAEA ETRS89) for 43 countries and territories in 2011. Intelligent dasymetric mapping (Mennis and Hultgren 2006) was employed in order to disaggregate best-available census data for each country (vector grids or census tracts) to built-up areas as mapped by the European Settlement Map 2016 (Ferri et al. 2014, Florczyk et al. 2016), and weighted by enhanced land use/cover data from a refined Corine Land Cover map where available (Freire and Halkia 2014). For eight countries, population grids were originally modeled at 10m spatial resolution and then aggregated to 100 m grid cells. This data product is freely accessible and downloadable at: http://data.jrc.ec.europa.eu/dataset/jrc-ghsl-ghs_pop_eurostat_europe_r2016a.

The U.S. Census Bureau has developed gridded **Demobase** population maps at 100m resolution, in the Geographic projection WGS84 for selected countries including Haiti, Pakistan, and Rwanda (e.g., Azar et al. 2010, Azar et al. 2013), as well as maps of subnational population by age and sex within administrative areas for various points in time since 1998. The Census Bureau has invested in efforts to provide data on population patterns by administrative areas and grid cells for various regions with a

5 focus on improving the availability of detailed population maps in regions likely in need of humanitarian relief and disaster assistance from external partners (U.S. Census Bureau 2018). Data inputs include census data from every country and territory that conducts a census, demographic surveys, maps of administrative boundaries from national and international mapping agencies, high- and medium-resolution satellite imagery, and a range of ancillary layers such as land cover, road networks, and elevation. Both Demobase gridded data and administrative-area based subnational datasets are freely accessible and

10 downloadable via links at: https://www.census.gov/programs-surveys/international-programs/about/global-mapping.html.

**4.3 Data availability**

The above described data products and their characteristics including the underlying population concept, method, resolution, points in time, the source for national-level population statistics used as well as reference links to access the data can be found in Table 2. All of these population grids are open access except two that have some restrictions. The different data producers

15 host the data in different ways, typically using internal servers and data repositories. Summaries and links to the various data repositories can be found at www.popgrid.org, facilitating access to, documentation and comparison of different data products. As mentioned before, the user can also find visual comparison tools (outputs as tables and through map services) that provide effective ways to perform visual analytics and identify differences in patterns of population distributions exhibited by the different data products.

**5. Different populations or different data? A Fitness-for-use perspective**

The process of creating gridded population products redistributes population estimates from census or administrative areas to grid cells, conditional on where human populations and settlements may be located. The nature, quality and accuracy of the input population data, the characteristics of the output gridded population dataset, the properties of the ancillary data used and

the implications of the methodological approach applied for population allocation and redistribution are all important determinants of spatial data quality in general (FGDC 1998, Guptill and Morrison 2013) but also help to shed light on the relative data quality of each of the population grids described in this review. While data quality and its reporting in standardized metadata has been the focus of much research in the last decades, the discussion of relative quality or fitness for use of spatial data has received less attention (see Devillers et al. 2007, Devillers et al. 2010, Ivánová et al. 2013). Since the described

population grids show fundamental differences, the question whether a data product is fit for a given purpose is of high relevance. Thus, in this section, we discuss several *determinants* (not an exhaustive list) that aid the data user in the assessment *of the data product's fitness for use* relative to the target application. We briefly discuss *data-related aspects* including scale, currency and semantics, as well as *modeling and processing-related implications* for uncertainty. We address them separately, but the reader may be reminded that all those relative quality aspects have to be understood interrelated as one can affect all

others. We will also address the problem of validation of large-scale population grids.

**5.1 Data aspects of relative quality**

The **accuracy** of the input census/population data and ancillary data includes thematic, spatial and temporal accuracies, which contribute to the level of uncertainty of the final data product. For this reason, the user needs to consider and understand what kind of data are input to a certain data production process. For census data, the completeness of coverage, the margin of error

(if sampled), the time period the census is taken and the positional accuracy of the boundaries are measures that can be used but might not be always known and the data need to be used with caution. This kind of knowledge is important to reflect when using population grids in a given region (e.g., Tatem 2014). With regard to the ancillary data, needless to say, the quality of the final population grid depends on the quality of the ancillary data used for redistributing population counts. Apart from the existence and strength of the assumed relationship between population and ancillary variables (Nieves et al. 2017), the accuracy

of these spatial layers themselves is critical for the accuracy of gridded population estimates. For example, the classification accuracy of built-up or developed land layers that are used to redistribute census counts to different regions tends to be lower in rural than in urban settings (Wickham et al. 2013, Leyk et al. 2014 and 2018, Uhl et al. 2018), but can also vary across larger regions and countries. The quality of remotely-sensed ancillary data also depends heavily on the characteristics of the instrument (optical daytime, optical night-time, or radar) and the processing algorithm (e.g., Small et al. 2005, Potere et al.

2009, Pesaresi et al. 2016b, Esch et al. 2017). Such differences propagate through to strongly influence the accuracy of the final population data product and may cause over- or underestimations in different subregions. Knowledge of such issues would be critical for the data user if population estimates in different regions are compared with each other. Due to the nature of the input and ancillary data, these accuracies translate into aspects of scale, currency and semantics critical for evaluating the fitness for use of the final population grids as discussed below.

**Scale**: Since input data are typically enumerated counts, issues due to spatial aggregation including the MAUP (Openshaw 1983), as the geographical manifestation of the ecological fallacy (Piantadosi et al. 1988, Waller and Gotway 2004), are one of the main sources of the "unknown." Differences in granularity of the input (census) data across different regions or countries must be taken into account since the same population redistribution model may perform very differently under different circumstances due to possible scale effects. In using the final population grids, the grid cell, defined by the spatial resolution

(that is, cell size; Table 2), would often be assumed to define the analytical scale (Montello 2001, Cao and Lam 1997). The user would often attempt to model a certain process or phenomenon of interest but often there is a mismatch between this 'operational' scale (e.g., Montello 2001, Maclaurin et al. 2015) and the analytical scale. However, it is imperative for the user

to understand that due to the difference between input population data (i.e., source unit) and output grid cell (i.e., target unit) granularity this assumption may be fundamentally flawed and result in either oversampling or generalization. For example, if the analysis is intended to be conducted at the neighborhood scale, population estimates provided in grid cells of 150m or 250m appear to represent meaningful target units. If these input data were at the census block or tract level, the grid cell size would represent an appropriate proxy and can be used as a valid analytical unit at the intended target scale. If, however, the input data originated from large administrative units (e.g., districts or county level source units) there would be a significant offset between input and output. In such cases, the user would face a higher risk of using oversampled population estimates that might result in higher degrees of local inaccuracy.

Creating equivalencies over time of finely resolved census units is challenging even in vector format; this problem is not necessarily abated when transforming vector data to grids. Differences in embeddedness of the population grid cells within census boundaries (when the census units are intrinsically larger than the average grid-cell size, also has implications for subsequent analysis using e.g., multi-level models over large areas and can become even more problematic if the census boundary – grid cell relationship changes over time thus impeding the creation of reliable population trajectories of place. To complicate matters, if ancillary data are used to redistribute population (e.g., to built-up portions of the source unit) based on existing relationships, such scale-related problems may be mitigated to some degree. In addition, variation in how a model is trained or the units selected to build the estimation model will influence the final gridded distribution (Sinha et al. 2019). For example, if census data from one region or country is very coarse, a model built based on finer-resolution data from a neighbouring region and then applied to the region of interest can be more accurate (Gaughan et al. 2015). Thus, scale effects are inherent to each of the described population grids at different degrees, and represent a geographically varying characteristic depending on the granularity of the input data, the strength of the associations between population and ancillary data, and the resolution of the output data. These effects need to be interpreted in the context of the target scale of the intended analysis and consideration should be given to the type of scaling needed to produce a given grid. For the interested reader, Ge et al. (2019) provides a comprehensive review on scaling considerations when working with geospatial Earth science data.

The **currency** of the data represents another important issue. In a few instances, underlying census data are old (e.g., in Haiti) or the period between censuses is more than 10 years. While some of the ancillary data are more or less constant over the near term (e.g., water bodies and permanent ice), there may also be temporal mismatches between population data and any of the intrinsically time-varying ancillary data layers (Figure 2). For example, it may be unknown whether a given built-up land grid cell has been developed at the time the census has been taken. Such temporal offsets may be critical if the assumption for the intended application necessitates a high degree of temporal agreement (currency). This form of uncertainty is difficult to handle and can be further complicated by differences across regions and countries. In response to this, few efforts (e.g., WorldPop and GHS-POP) ensure the use of temporally implicit or invariant ancillary data in the modeling process (Gaughan et al. 2016) (Table 1). However, even under those conditions, there might still be underlying issues for projecting forward/backward from census data for a target year of interest. The user is well advised to understand the gridded population estimates as approximations over a period of time and avoid flawed assumptions of high currency in a given analysis.

**Semantics**: As mentioned before, what the population modeled represents can be very different among data products. This meaning can even be different within one product if, e.g., the census input data account for different population concepts or population groups in different regions or countries. For example, the population estimate might refer to *de-jure* (or legal) populations vs. *de-facto* (or present) populations and using the one over the other product would possibly result in dramatically different results. The user has to be aware that data on resident populations as provided by censuses is itself a convention, whose distribution never occurs at any moment in time (*de jure* census population) or if it does occur (*de facto*, location at the time of the census) that distribution may not be representative of a different situation or in the medium/long term (i.e., a year): the concept of usual residence. Most of the global population data products use a night-time / usual residence (*de jure*) concept, or mostly rely on underlying data that use a *de jure* concept, with LandScan being the notable exception. Thus, the user is well

advised to be aware of the meaning of the populations modeled in the population grid in question to avoid such misinterpretations, as indicated in Table 2. The aspects of scale mismatch described above can further add to semantic differences since due to such aggregation effects, different populations may be modeled. Thus, these implications have to be understood by the user, spatially and semantically, and caution is advised when interpreting analytical results.

## 5.2 Processing- and model-related implications of uncertainty

Regardless of the approach of choice employed for data production, all efforts described in this review do carry out some form of data conversion (e.g., vector-to-raster) and data integration (re-allocation or resampling). The different population grids described are based on varying levels of modelling intensity (unmodeled, lightly and highly modelled) as indicated in Table 2. However, any such data processing step will propagate uncertainty in some way and have consequences for the quality of the outcome data and the subsequent analyses, depending on the input data quality as described above. For example, if large census units (e.g., counties or districts) in vector format are converted to grid cells (rasterization) of fine spatial resolution (e.g., 150m), while there is a clear scale effect to be addressed (see above), the resulting population estimates may differ dramatically for different redistribution models applied that may or may not use ancillary data. The data user needs to be aware that existing uncertainty is not eliminated by applying certain models or integrating different data sources. However, through the process of data integration we may be able to improve the accuracy based on spatial refinement strategies such as dasymetric modeling (Mennis 2009).

GPW, GHS-POP and HYDE do not employ statistical methods to produce their grids, and thus traditional metrics of uncertainty are not available. Because fine resolution inputs reduce errors of aggregation, GPW reports the number of input units per country used in the gridding process. Nevertheless, errors may persist in countries with highly variable input units. For example, Sahelian countries have finely resolved units for densely populated areas but very coarse units for sparsely populated regions.

The specific model applied to re-allocate population counts and densities, which can be empirical or statistical, will always have some error. This error relates to the estimated relationship between population and ancillary variable and not to the final population estimate which also may incorporate uncertainty due to error in the input population data or ancillary data. When the modelling process is statistical or hybrid such as in the case of the WorldPop, Landscan and WPE, estimates of such model errors (e.g., standard error of regression coefficients, prediction error) can be derived as a by-product of the modeling process. To fully understand the quality of a population grid, the error of the applied model needs to be evaluated. Highly accurate ancillary data are not useful if the relationship to population is weak or the model applied inappropriate, and thus the model predictions are unreliable (e.g., low R squared, or deviance explained). Such prediction errors are often assessed in comparison to alternative models but are hard to quantify in the absence of validation data. To complicate matters, the same model might perform very differently in different geographies or under different environmental conditions, an effect known as spatial non-stationarity or spatial variation of the target relationships (Fotheringham et al. 1996). Such variations will further affect the model predictions if left unaccounted.

## 5.3 Validation challenges

Validation of population data has always been a challenge, simply because validation data at fine resolutions are rarely available and even when available, may exist at different time periods or confidentiality rules may limit their use in order to not expose individual and household level information. Access to such confidential data is only possible with special permission or sworn status and even then, often the demographic data are only a sample of the whole population. These challenges can be very different between countries and thus a validation that may be possible in one country does not necessarily translate to another location. For example, Tiecke et al. (2017) compare the locations of the population grid to the GPS coordinates of the nationally representative sample of almost 12,000 households interviewed for a survey in Malawi.

While a true validation of the gridded output remains a challenge, it is possible to internally test the accuracy of the modeling approaches (Gaughan et al. 2015, Sorichetta et al. 2015, Reed et al. 2018). Such an assessment can be done when different levels of census input data are available for use in a model. The approach leverages the coarser level data in different modelling approaches and then compares the gridded outputs to the finer level census data to determine how well and plausible populations were distributed.

Validating ancillary data may have its own challenges. However, the existence of new, more detailed reference data in some regions (e.g., parcel data, crowdsourcing data) has helped to make progress in evaluating land cover data and built-up land layers, which is key to most of the described population grids (See et al. 2015, Leyk et al. 2018, Leyk and Uhl 2018). In general, depending on the level of land development and land use patterns in the region of interest, different products may serve the intended purpose differently.

## 6. Fit for use or not fit? Concluding remarks and future work

The different critical elements described above all have some impact on the fitness for use as a measure of relative data quality. Despite the importance of data quality, it does not receive the attention it deserves, in part because comparative measures may be difficult to conceive, derive or quantify. Furthermore, such assessment also importantly depends on the application of interest. The different aspects above have to be seen in context and considered interrelated. Different analytical and data processing steps such as conversions or data integration do not cause isolated uncertainties but through all those steps uncertainty can be propagated and thus becomes difficult to control and account for.

Whether or not data are fit for an intended use is not based on standardized measures nor is it well understood as to what the concept of fitness for use actually entails. Often it is at the discretion of the data user to decide whether the use of a given data product is appropriate or not, particularly in the age of open public data and open science. Based on the above discussion, there are a few guidelines that, in general, can help a user make informed decisions related to the fitness of a given population data product for their intended use:

**(1) How important is spatial refinement of the population grid to be used?** In the last 20 years, considerable attention has been paid to the spatial refinement of gridded population estimates. Some applications such as estimation of populations at risk of seaward natural hazards benefit substantially from these improvements. Other applications such as some climate scenario modeling do not require such finely resolved data as information on the general spatial distribution of population at moderate resolution would be sufficient.

**(2) Does the analysis focus on urban populations?** Closely related to the above concern, if the aim of the analysis is to examine urban population distribution as opposed to rural population, one would be better off using a data set for which information on human settlements or urban extents (e.g., in GHS-POP and GRUMP) has been used in the modeling. Urban land tends to be concentrated and can be clearly distinguished from the surrounding areas in remote sensing images and thus settlement data products (or other measures of urban extent) are effective in spatially refining population data along an urban gradient that will most likely improve the spatial precision of resulting estimates (though there is always the possibility that they will over-concentrate population in built-up locations). In contrast, data products that do not include such refinements tend to underestimate urban population. Data with extremely high resolution may mitigate such effects even if no settlement data were involved in the data production.

**(3) What is the target population for the question at hand?** Questions aimed at understanding long-term population change are likely to be well served by the use of population grids that represent night-time, residential population. In other instances, however, for example on emergency response, one may need to know where populations are likely to be during the day-time or rely on an ambient population concept and thus would be better served by data products that incorporate that concept in the modeling process.

**(4) Is the population grid being used to model other outcomes?** If so, and if that outcome may be one of the ancillary variables (or one closely linked to it) used in the production of the population grid, one needs to select a population grid that is not endogenous to the question at hand. For example, if the goal of the analysis is to estimate changes in built-up area using population as one of the explanatory variables, then one must not use GHS-POP (which uses built-up area). This necessitates that users become familiar with the ancillary data used in the production of the population grids, and even those used, perhaps, in training data sets but not actually the modelling.

**(5) Analysing change over time?** If one's goal is to examine change over time in population distributions, one of the data products representing multiple years is most suitable. However, there may be differences in how different grids have been generated for specific years. In order to analyze change in population distributions, ideally a data product built from data layers representing the respective time period would be preferred. Data which represents only one time period and applies, say, national-level growth rates to derive data distributions for earlier time periods, would be less amenable.

**(6) How have these data sets been used previously?** Some of the data providers make available citation lists of publications from the providers' team or the broader user community that may provide some guidance for the novice user. For example, GPW, WorldPop, LandScan, and others provide such lists which are extremely useful as a collection of common applications in which those data have been used. Based on such lists, the user can explore whether prior use of the data products appears to be appropriate with regard to target applications and how these scenarios compare to their own study. These data sets are used in combination with other spatially rendered data, whether those data are thematically environmental, health-related or social in nature, leading to a wide array of usages. However, they are typically not combined with other data that are limited in their spatial specificity (such as historic census tables or national-level survey data).

This review is an attempt to shed light on underlying data considerations to raise the awareness of relative data quality concerns related to the described population data products. The data user community is encouraged to consider the described quality aspects and metadata carefully, before making decisions about any given data product's fitness for the target application. This can include the full assessment of the above aspects, the use of metadata as well as sensitivity analysis including running an analysis at different spatial resolutions or the comparison of analytical results using different population grids (see e.g., Mondal and Tatem 2012; Tatem et al. 2011) to understand and quantify the sensitivity of the study results.

There has been significant progress in the spatial rendering of population and related characteristics in the past 20 years, but persistent challenges remain. We depend on existing population grids that are created using ancillary data to provide hints for where people live or spend time. In an ideal world, the research community would also have access to detailed building footprint and height data for all structures, and know whether these structures are residential or commercial, if indeed they are occupied at all to pair with population data. Future work will help to close these gaps by employing new high-resolution satellite technology as well as more reliable population surveys. This includes new and improved nighttime lights products (e.g. Visible Infrared Imaging Radiometer Suite (VIIRS) with respect to DMSP-OLS Nighttime Lights), that have been already successfully tested in urban mapping applications (Elvidge et al. 2017), and settlement data production (GHSL, GUF, as well as at Digital Globe, etc.) to further refine the available population grids.

It is important to note that since the first global population grid, the emphasis has been on producing grids of population counts and density rather than any other population attribute. While this emphasis has its obvious roots in the importance of population as a denominator, it also arises from the simple constraint that population is the most consistently-measured variable across place and time (though not without historical exception). This community should accept it as a challenge to expand into other population attributes in the near future. Members of the POPGRID data collaborative are investing work in a number of emerging areas, including future population projections (Jones and O'Neill 2016), population projections incorporating climate change impacts (Rigaud et al. 2018), near-real-time population modeling (Bharti et al. 2015), mobility mapping, population dynamics (Deville et al. 2014), increased temporal resolution (Batista et al. 2018) and working directly with national statistical

offices to improve the spatial accuracy of census products (www.grid3.org). These efforts often make use of novel data streams such as cell phone call detail records or social media data, or best practices in data collection using mobile devices. Finally, future work will be dedicated to improving the accuracy of population estimates, particularly in rural regions, where the reliability of existing data products is limited to date.

## Author contribution

SL, AG, SA, AdS, DB, SF, AT, MP, KM and ML conceptualized this review article and formulated its vision. SL, AG, SA, DB and AdS structured the manuscript and developed contents of the various sections. SL, in collaboration with AG, SA and AdS, drafted the manuscript, with contributions from all co-authors. DB, AR, FRS, BB, CF, JC, AS, SF, MP and KM wrote and revised data product descriptions and provided insights on data product characteristics and underlying procedures. All authors have read and revised the manuscript. AG and LP created the figures.

## Competing interests

The authors declare that they have no conflict of interest.

## Acknowledgements

15  POPGRID has been supported by seed funding from the Columbia University Earth Institute's Cross-Cutting Initiatives and the Bill & Melinda Gates Foundation. It is an element of the Group on Earth Observations (GEO) Human Planet Initiative (HPI) and is exploring linkages with key sustainable development data organizations and networks. SL is supported by the Eunice Kennedy Shriver National Institute of Child Health & Human Development of the National Institutes of Health under Award Number P2CHD066613. The content is solely the responsibility of the authors and does not necessarily represent the official views of the National Institutes of Health. SL and DB are supported by the US National Science Foundation award #1416860 to the City University of New York, the Population Council, the National Center for Atmospheric Research and the University of Colorado at Boulder. AEG, FRS, AS and AJT are supported through funding from the Bill & Melinda Gates Foundation Investment ID OPP1134076, and SA and AD by the Gates Foundation Investment ID OPP1177328. AEG and FRS are also supported through the NASA Land Cover and Land Use Change Program and the NASA GEO-Human Planet Program. We are grateful to the many individuals who produce, test and distribute these invaluable databases.

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
