# Peer review of "The spatial allocation of population: A review of large-scale gridded population data products and their fitness for use"

_Earth System Science Data, 2019_

## Referee Comment (RC1) · Tracy Kugler (Referee) · 5 Jul 2019

This article presents a valuable summary and comparison of the major gridded population data products currently available. While the datasets discussed are generally well-established and have each been individually described in previous publications, having a summary comparing the key properties of these datasets in one place will be very useful for researchers trying to determine which product is best suited for their particular application. Furthermore, the concept of "fitness for use" is a helpful approach in this context. Each of the datasets discussed incorporates different types of input information and applies different methodology, which results in the final data products

having noticeably different properties. In the absence of definitive means to validate
and assess the accuracy of gridded population data products (as discussed in the article), researchers should thoughtfully consider how the properties of a given dataset
will affect the results of their application. The guidelines presented in the final section
of this article provide a helpful framework for thinking through such considerations.

In general, the article is well-written, and the concepts are presented clearly. I have
just one suggestion about potential additional content. I have some suggestions about
rearranging the structure of the article to clarify the flow of ideas. I also have some more
specific requests for clarification, suggestions regarding the figures, and one technical
issue.

__Content suggestion__ Could the authors comment on how total population grids
may be combined with data (e.g. published census tables) containing a wider range of
characteristics?

__Structural Suggestions__

* Rearrangement of paper sections I suggest moving the Review of current data products (sec. 3) to follow the discussion of methods. Having the methods descriptions first
would give readers better context to understand the methods used in each data product. Additionally, within the Key methods and ancillary data section (sec. 4), I suggest
swapping the order of the Ancillary data (4.1) and Methods for population redistribution
(4.2) subsections. Discussing different types of ancillary data makes more sense once
the user knows what the ancillary data are used for. Conversely, the specifics of ancillary data are not necessary for understanding the methods. With this arrangement,
the discussion starting at line 5 on p. 15 could be set off as its own section (headed
something like, "Multiple answers and uncertainty") and placed after the discussion of
ancillary data. As a central concern of the article, this discussion seems to deserve its
own subsection. Following these suggestions would result in the following structure for
the paper: * Introduction (unaffected) * Background and historical development (unaffected) * Key methods and ancillary data (moved and rearranged) * Methods for population redistribution * Ancillary data * Multiple answers and uncertainty * Current data products (moved) * A fitness for use perspective (unaffected) * Concluding remarks and future work (unaffected) This structure puts the discussion of Methods for population distribution immediately following the Background and historical development section, which seems like a natural progression.

* Review of Current data products Within this section, the description of each product should be as parallel as possible. e.g., resolution, coordinate system, and type of population mapped in first sentence. Then narrative overview of input data and methods. Concluding with availability in the last sentence. Currently, the descriptions of Esri WPE, WorldPop, HYDE, and Demobase stray from this structure. For national and regional/continental products, coverage (countries/continent/region included) should be stated early in the description (first or second sentence). If possible, it would also be helpful to have coverage information for these products in table 1.

\_\_Specific Clarification Questions\_\_

* p. 9, line 16: What is the basis for HYDE prior to 1950? * p. 14, lines 18-23: It is not clear whether or how the statistical modeling approaches that are the subject of this paragraph differ from statistical approaches to dasymetric allocation mentioned on line 14. Please clarify. * p. 14, line 24 - p. 15, line 4: Again, it is not clear to me how so-called hybrid approaches fundamentally differ from non-hybrid weighted dasymetric methods. This brief paragraph doesn't really help clear up my confusion. Is the difference primarily in the techniques used to generate the weights? If so, what exactly is it that differentiates a hybrid approach from a straight weighted dasymetric approach? * p. 19, lines 30-31: It is somewhat confusing to draw a connection between de-jure and night time and de-facto and daytime populations. In the context of census data, de-jure and de-facto typically refer to longer timescales than day vs. night. e.g., If a person typically lives at a given address but is away on vacation on the day of the census, they could be counted at that location in a de-jure census, but at the location of
their vacation in a de-facto census. If a person commutes work (their daytime location) on the day of the census, though, they would still be counted at their residence, even in a de-facto census. * p. 20, lines 16-19: Maybe include a note of caution that the statistically estimated error relates specifically to the estimated relationship between population and ancillary data, not directly to the final population estimate (which also may incorporate uncertainty due to uncertainty in the input population and ancillary data themselves). * p. 21, line 18: Other questions are phrased in relation to the analysis under consideration. To be consistent, question 2 could be phrased as something like, "Does the analysis focus on urban populations?"

__Suggestions on Figures__

* Figure 1: As presented, this figure is somewhat difficult to interpret. * The specific "Good pixel proportion" of each Landsat scene seems like more detail than is necessary for this figure. The cost of introducing the rather confusing vertical axis in the top part of the figure to convey this detail doesn't seem worthwhile. * It seems odd to tie OSM to a specific point in time, since it is continuously evolving. * If a main intention of the figure is to compare the time points of ancillary data to census data, the census markers should be drawn out to the edge of the figure, rather than buried in the middle.

* Figure 2 * Label the 4 quadrants of the figure as a,b,c,d, and refer to specific quadrant(s) in the text to clarify the link between the text description and the illustration of a particular method. (Specifically, are the three methods mentioned on lines 13 & 14 of p. 14 – binary, 'intelligent', and statistical – meant to relate to the three variations of dasymetric illustrated in the figure?) * Indicate within the figure that the 6 gray cells represent built-up area or a similar category as shown in ancillary data

* Figure 3: A more intuitive visual arrangement of this figure would be to have panel (a) (input population data) on its own row at the top, followed by panels (b-e) (ancillary data) on two rows, followed by panel (f) (output population grid) on its own row at the bottom.

\_\_Technical issue\_\_ In the PDF download of the manuscript, clicking on the link in the caption for Table 1 attempts to go to: https://www.popgrid.org/popgrid_files/popgrid-5, which doesn't exist. (The '5' is from the line number on the page.) The URL as written in the manuscript does work. It might be best just to link to the HTML version at https://www.popgrid.org/compare-data, which is consistent with the link on p. 12 (line 26)

―――――――――――――――――――――

---

## Referee Comment (RC2) · Stephen Matthews (Referee) · 8 Jul 2019

Thank you for the opportunity to read and comment on the manuscript "Allocating People to Pixels: A Review of large-scale Gridded Population Data Products and their Fitness for Use."

First and foremost this is a valuable resource/summary and is most welcome. I haven't used many of the products in my own research but in my instructional role I have referred to these products and encouraged students to explore them. This comparative summary provides a useful overview to the data products, a set of spatial-temporal and modeling issues, and introduces guidelines to help determine fitness of use. That is, I

can envision promoting this "review" in instruction and reference to colleagues. I liked the paper.

My comments are general and are provided to help potentially improve the experience for the reader. They are not listed in any priority order but just things that occurred to me as I read the material.

1: Why use the word "pixels" in the title? ... and not "grids"? While the journal readership will be familiar with both terms (and I know there are publications on people and pixels) but all of the products highlighted are gridded population data sets. Further, the abstract does not contain the word pixel(s) and the word is rarely used in the manuscript (just in a few subheadings). Even in subheadings I would prefer to see 'grids.' Then why heading #4 "people in places"?

2: In section #3 the reader is introduced to the POPGRID website at www.popgrid.org. This is great. Many of the tables in the manuscript are from this website but curiously they are either selected extracts or restructured. One would have thought that consistency between the two would be more useful and that the paper should follow the format of the tables that are online. This is especially so as in Table 1 where there is no obvious structure to the listing of data resources but the same table online organizes these same data resources/products by whether the data are "unmodeled," "lightly modeled," or "highly modeled." I would suggest that the latter helps the reader and this is especially so when it comes to later sections of the paper that introduce the various methods for population redistribution (and also section #5.2)

3: I would much prefer that Table #1 comes before the start of section 3.1. It has been introduced but we wait for until after the short description of all component data resources before we encounter it.

3b: Table #1 should be titled "Detailed characteristics and availability ... " as it covers "availability" as well.

3c: some columns in Table #1 are closely related to "fitness for use" expanded on in section #6 and alludes to at other points in the paper, so perhaps highlight them as such in the table and/or more explicitly refer back to them in the text (perhaps both when first introduced and in sections #5 and #6). This would help tie the two parts of the paper together (i.e., the "review" and the "fitness of use" sections).

4: I agree that HYDE is a very interesting data set and perhaps of import to the readers of the journal but these data are unique historically and also are available at a fairly crude level compared to all other gridded data products. That is, is this product sufficiently different to include as say supplemental material rather than list with the others?

5: I wondered if the ancillary data section (#4.1) and Figure 1 should come after section 4.2. Perhaps a relatively minor issue.

5b: Figure #1 has a lot of information that might be better explained. Ditto Figure #2.

6: I know the readership of the journal is likely to be very sophisticated in this area but I still think that a glossary of key terms is necessary and should be presented early in the paper. Maybe I am thinking about my role as an instructor but I suspect in assigning this reading I would have to prepare the novice/intermediate user to several key terms.

7: I very much liked Figures 3 and 4 and their brief description.

8: I wondered if lines 11-15 could be bolded/italicized or perhaps just a separate paragraph to give emphasis to the interrelatedness of the determinants of fitness for use.

9: Fitness of use #2 focuses on urban population analysis. This seemed like an opportunity to link to Table 4 at www.popgrid.org/compare-data on "Global and Continental Urban Extent / Settlement Layers: Summary Characteristics."

10: Not sure there is, but if there was, an expansion of the theme "How have these data sets been used previously" would be useful. Some good examples, just wondering about classic papers or studies that could be used as exemplars.

11: I recognize that there are pieces of the next comment scattered in the paper but for the user I wondered if more could be made of time-constant geographies and issues related to the embeddedness of units of analysis and scale. Some readers will be interested in trajectories of place as well as multilevel modeling (perhaps) but the number of temporal data collection points and the embeddedness of levels is not always made explicit. I agree that this can be complex but alluding to these issues in the context of research questions can help in the fitness of use.

12: While not the main purpose I wasn't sure about all the referencing to social media. I wonder if the trap of thinking about the future has compelled the authors to discuss this but it all reads fairly superficially. At least to me. I do think an expanded section on data challenges (e.g., non-representative samples) and fast changing data environments (social network data, real-time data) are worth discussing but if so, then in more depth.

13: Minor – gridded data sets are not particularly recent and they are part of the history and formation of remote sensing, raster GIS, and map algebra tools/perspectives in the spatial sciences (in disciplines spanning the environmental, geographic, and social sciences). An early citation to time-constant geographies and socioeconomic applications in GIS that focused on grid-based data products include David Martin (1991, 1995), now at Southampton.
* * *

---

## Author Comment (AC1) · 21 Jul 2019

**Responses* to comments made by Reviewer 1 (Tracy Kugler):**

This article presents a valuable summary and comparison of the major gridded population data products currently available. While the datasets discussed are generally well-established and have each been individually described in previous publications, having a summary comparing the key properties of these datasets in one place will be very useful for researchers trying to determine which product is best suited for their particular application. Furthermore, the concept of "fitness for use" is a helpful approach in this context. Each of the datasets discussed incorporates different types of input information and applies different methodology, which results in the final data products having noticeably different properties. In the absence of definitive means to validate and assess the accuracy of gridded population data products (as discussed in the article), researchers should thoughtfully consider how the properties of a given dataset will affect the results of their application. The guidelines presented in the final section of this article provide a helpful framework for thinking through such considerations. In general, the article is well-written, and the concepts are presented clearly. I have just one suggestion about potential additional content. I have some suggestions about rearranging the structure of the article to clarify the flow of ideas. I also have some more specific requests for clarification, suggestions regarding the figures, and one technical issue.

*RESPONSE: Thank you very much for the positive evaluation and the constructive comments and suggestions. Our responses to the suggestions and comments are found beneath yours below.*

__Content suggestion__ Could the authors comment on how total population grids may be combined with data (e.g. published census tables) containing a wider range of characteristics?

*RESPONSE: This is a great question! We added additional commentary that population counts and densities are the common variables considered, with a challenge to expand this in the future in the concluding remarks. However, we also include a reminder that these data are designed to be used with spatial-rendered data, thus making linkages to some census tables and national-level survey data, for example, challenging (page 22, lines 16-19 and lines 36-40). In our opinion, while there is much potential for using these grids to refine other census attributes, supply denominators and many other usages, we felt that topic is worthy of its own consideration beyond the scope of this paper. One data service (ESRI's GeoEnrichment Service) has implemented some algorithms to estimate other census attributes based on the population distribution, but these procedures are not yet fully implemented nor validated and due to the underlying complexity we decided that this additional topic could not be given full attention.*

__Structural Suggestions__: * Rearrangement of paper sections

I suggest moving the Review of current data products (sec. 3) to follow the discussion of methods. Having the methods descriptions first would give readers better context to understand the methods used in each data product. Additionally, within the Key methods and ancillary data section (sec. 4), I suggest swapping the order of the Ancillary data (4.1) and Methods for population redistribution (4.2) subsections. Discussing different types of ancillary data makes more sense once the user knows what the ancillary data are used for. Conversely, the specifics of ancillary data are not necessary for understanding the methods. With this arrangement, the discussion starting at line 5 on p. 15 could be set off as its own section (headed something like, "Multiple answers and uncertainty") and placed after the discussion of ancillary data. As a central concern of the article, this discussion seems to deserve its own subsection. Following these suggestions would result in the following structure for the paper: * Introduction (unaffected) * Background and historical development (unaffected) * Key methods and ancillary data (moved and rearranged) * Methods for population redistribution * Ancillary data * Multiple answers and uncertainty * Current data products (moved) *A fitness for use perspective (unaffected) * Concluding remarks and future work (unaffected). This structure puts the discussion of Methods for population distribution immediately following the Background and historical development section, which seems like a natural progression.

*RESPONSE: Thank you for those suggestions. We agree that the flow of the paper could be improved through these structural modifications. We restructured the paper as follows:*
*\* Introduction*
*\* Background and historical development*
*\* Key methods and ancillary data*
  *-- Methods for population redistribution*
  *-- Ancillary data*
  *-- Different methods and sources of uncertainty*

*\* Current data products*
*\* A fitness for use perspective*
*\* Concluding remarks and future work*

*Figures 1 and 2 are now Figures 2 and 1, respectively; Tables 1 and 2 are now Tables 2 and 1, respectively, and transitions between sections have been adjusted.*

\* Review of Current data products Within this section, the description of each product should be as parallel as possible. e.g., resolution, coordinate system, and type of population mapped in first sentence. Then narrative overview of input data and methods. Concluding with availability in the last sentence. Currently, the descriptions of Esri WPE, WorldPop, HYDE, and Demobase stray from this structure. For national and regional/continental products, coverage (countries/continent/region included) should be stated early in the description (first or second sentence). If possible, it would also be helpful to have coverage information for these products in table 1.
**RESPONSE***: We improved the synchronization of the presentation of the different data products in "Current data products, characteristics and availability" (now Section 4) to ensure that the different characteristics are described in the same order to facilitate readability and cross-comparison.*
*We also ensured all regional/national data product descriptions include the geographic coverage. We prefer not to include those coverages in Table 1 (now Table 2) since it already contains a lot of information, and we assume that coverage will change over time as the data producing institutions continue their efforts.*

**__Specific Clarification Questions__:**
\* p. 9, line 16: What is the basis for HYDE prior to 1950?
**RESPONSE***: We added text about the pre-1950 bases for HYDE: "For the period prior to 1950, historical input population estimates were taken from the general literature and supplemented with the sub-national population numbers and country-specific sources to build time series for each province or state of every country" (page 16, lines 11-13).*

\* p. 14, lines 18-23: It is not clear whether or how the statistical modeling approaches that are the subject of this paragraph differ from statistical approaches to dasymetric allocation mentioned on line 14. Please clarify. \* p. 14, line 24 - p. 15, line 4: Again, it is not clear to me how so-called hybrid approaches fundamentally differ from non-hybrid weighted dasymetric methods. This brief paragraph doesn't really help clear up my confusion. Is the difference primarily in the techniques used to generate the weights? If so, what exactly is it that differentiates a hybrid approach from a straight weighted dasymetric approach?
**RESPONSE***: We agree that the differentiation between the different approaches needs to be clearer. We revised these methodological descriptions to differentiate between traditional dasymetric mapping, statistical approaches and hybrid approaches. In this revised text we explicitly point out that statistical approaches rely on statistical estimation of relationships used for population modeling while traditional approaches mostly rely on presence/absence rules or empirically derived weights. Furthermore, the hybrid approaches differ from statistical ones in that there is a more direct link between statistical analysis (often machine learning; to estimate population weights) and subsequent dasymetric redistribution using those population weights (see page 8, lines 12-33).*

\* p. 19, lines 30-31: It is somewhat confusing to draw a connection between de-jure and night time and de-facto and daytime populations. In the context of census data, de-jure and de-facto typically refer to longer timescales than day vs. night. e.g., If a person typically lives at a given address but is away on vacation on the day of the census, they could be counted at that location in a de-jure census, but at the location of their vacation in a de-facto census. If a person commutes work (their daytime location) on the day of the census, though, they would still be counted at their residence, even in a de-facto census.
**RESPONSE***: We agree on this point and removed the links between these concepts in relevant places.*

\* p. 20, lines 16-19: Maybe include a note of caution that the statistically estimated error relates specifically to the estimated relationship between population and ancillary data, not directly to the final population estimate (which also may incorporate uncertainty due to uncertainty in the input population and ancillary data themselves).

*RESPONSE: We revised the text to clarify this point: "This error relates to the estimated relationship between population and ancillary variable and not to the final population estimate which also may incorporate uncertainty due to error in the input population data or ancillary data" (page 20, lines 23-24).*

\* p. 21, line 18: Other questions are phrased in relation to the analysis under consideration. To be consistent, question 2 could be phrased as something like, "Does the analysis focus on urban populations?"
*RESPONSE: We agree and rephrased this question accordingly as suggested (page 21, line 28).*

__Suggestions on Figures__
\* Figure 1: As presented, this figure is somewhat difficult to interpret. \* The specific "Good pixel proportion" of each Landsat scene seems like more detail than is necessary for this figure. The cost of introducing the rather confusing vertical axis in the top part of the figure to convey this detail doesn't seem worthwhile. \* It seems odd to tie OSM to a specific point in time, since it is continuously evolving. \* If a main intention of the figure is to compare the time points of ancillary data to census data, the census markers should be drawn out to the edge of the figure, rather than buried in the middle.
*RESPONSE: We revised the figure (now Figure 2, page 9) to address the above concerns and with the goal to make this figure more readable and embedded into the main text. We decided to leave the beam symbolizing the acquisition dates of satellite data in the figure but changed and simplified the layout. Furthermore, we edited the visual presentation of the ancillary variables to better reflect on their availability and temporal offsets. We also revised the figure caption to emphasize that the purpose is to show the temporal offsets between ancillary and input data as well as explain the reason why the source remote sensing data capture dates are shown. Finally, we revised the main text to better refer to the figure and embed it into the descriptions of ancillary data (e.g., page 9, lines 5-7, and references in different places). See figure 2 and caption below:*

[Figure]

*Figure 2. Identification of different ancillary data that inform spatial and temporal interpolation approaches to create gridded population data across scales of interest. Temporal fidelity in the Landsat (30m resolution; with varying proportions of cloud-free area) and MODIS (250m resolution) sensors are shown in relation to typical points in time for censuses alongside several derived ancillary data products such as the European Space Agency (ESA) annual land cover data (300m resolution), and the Global Human Settlement Layer (38m resolution) at various publication dates. The Global Urban Footprint (GUF+) exists for one point in time only. Also noted are OpenStreetMap data, vector-based information that is increasingly explored as a possible ancillary data source, which can be acquired anytime and is potentially useful for more contemporary time periods as a static variable; as it is continually evolving, it's currency may deviate by region.*

\* Figure 2 \* Label the 4 quadrants of the figure as a,b,c,d, and refer to specific quadrant(s) in the text to clarify the link between the text description and the illustration of a particular method. (Specifically, are the three methods mentioned on lines 13 & 14 of p. 14 – binary, 'intelligent', and statistical – meant to relate to the three variations of

dasymetric illustrated in the figure?) * Indicate within the figure that the 6 gray cells represent built-up area or a similar category as shown in ancillary data

*RESPONSE: We also revised this figure (now Figure 1, page 7) according to some of these suggestions but also improved the general design to better label the different forms of interpolation approaches. We also included labels a-d and use them in the main text to better link between text and figure (page 7, line 11 and page 8, lines 14-20). See figure and caption below:*

[Figure]

*Figure 1. Schematic illustration of different types of techniques for population redistribution or allocation from source to target grid cells: (a) Areal weighting as the simplest form of areal interpolation that does not use any ancillary variables; (b) Dasymetric mapping using binary ancillary variables that inform and refine areal weighting; (c) Dasymetric mapping using varying population weights that may be empirically derived or based on set rules; (d) Statistical modelling to estimate relationships that can be used for population modelling. The different grey tones in (b)-(d) indicate different underlying data informing the areal interpolation process.*

* Figure 3: A more intuitive visual arrangement of this figure would be to have panel (a) (input population data) on its own row at the top, followed by panels (b-e) (ancillary data) on two rows, followed by panel (f) (output population grid) on its own row at the bottom.

*RESPONSE: We understand the idea of re-arranging this figure to improve readability of input and final output. However, after testing some options we concluded this current format and arrangement remains the most optimal in terms of space use and comparability. Thus we decided to leave this figure in its current state but made some minor improvements to the individual panels.*

__Technical issue__

In the PDF download of the manuscript, clicking on the link in the caption for Table 1 attempts to go to: https://www.popgrid.org/popgrid_files/popgrid-5, which doesn't exist. (The '5' is from the line number on the page.) The URL as written in the manuscript does work. It might be best just to link to the HTML version at https://www.popgrid.org/compare-data, which is consistent with the link on p. 12 (line 26)

*RESPONSE: We were able to reproduce this issue, and will work with the editorial office to fix this link. It appears the link is corrupted when converting to a PDF.*

**Responses to comments made by Reviewer 2 (Stephen Matthews):**

Thank you for the opportunity to read and comment on the manuscript "Allocating People to Pixels: A Review of large-scale Gridded Population Data Products and their Fitness for Use." First and foremost this is a valuable resource/summary and is most welcome. I haven't used many of the products in my own research but in my instructional role I have referred to these products and encouraged students to explore them. This comparative summary provides a useful overview to the data products, a set of spatial-temporal and modeling issues, and introduces guidelines to help determine fitness of use. That is, I can envision promoting this "review" in instruction and reference to colleagues. I liked the paper. My comments are general and are provided to help potentially improve the experience for the reader. They are not listed in any priority order but just things that occurred to me as I read the material.

*RESPONSE: Thank you very much for the support and positive evaluation as well as the constructive comments and suggestions, to which our replies are found below.*

1: Why use the word "pixels" in the title? ... and not "grids"? While the journal readership will be familiar with both terms (and I know there are publications on people and pixels) but all of the products highlighted are gridded population data sets. Further, the abstract does not contain the word pixel(s) and the word is rarely used in the manuscript (just in a few subheadings). Even in subheadings I would prefer to see 'grids.' Then why heading #4 "people in places"?

*RESPONSE: We considered the term pixel synonymous to grid cell. But we appreciate this comment and agree that pixel is less appropriate since grid cell and grids are mainly used throughout the manuscript and better connect to the target communities. We changed pixels to grids or grid cells as appropriate. For example, we adopted the main title to: "The spatial allocation of population: A review of large-scale gridded population data products and their fitness for use", and changed Section 2 heading to "People as gridded distribution" (page 5, line 35).*

2: In section #3 the reader is introduced to the POPGRID website at www.popgrid.org. This is great. Many of the tables in the manuscript are from this website but curiously they are either selected extracts or restructured. One would have thought that consistency between the two would be more useful and that the paper should follow the format of the tables that are online. This is especially so as in Table 1 where there is no obvious structure to the listing of data resources but the same table online organizes these same data resources/products by whether the data are "unmodeled," "lightly modeled," or "highly modeled." I would suggest that the latter helps the reader and this is especially so when it comes to later sections of the paper that introduces the various methods for population redistribution (and also section #5.2)

*RESPONSE: Thanks for pointing this out. The Table (now Table 2, pages 13/14) was intended to present a shorter version or synthesis of the POPGRID tables found through this link. We changed the caption for this table by adding "More information about these and other data products can be found at…" to make sure the reader will be able to find more detailed descriptions delivered through popgrid.org that are expected to continually evolve. But we agree with the argument to include "unmodeled", "lightly modeled" and "highly modeled" since this is indeed very useful for the fitness-for-use discussion. Thus we inserted these labels in the table as a left-side vertical aligned column for the global population grids. We also refer to these labels in the data product description (Section 4, page 13, lines 7-8 and individual product summaries in Section 4.1) as well as in the Discussion section (Section 5.2, page 20, lines 7-9).*

3: I would much prefer that Table #1 comes before the start of section 3.1. It has been introduced but we wait for until after the short description of all component data resources before we encounter it.

*RESPONSE: We agree with this critique and moved the table (now Table 2, pages 13/14) right before this section (now Section 4.1).*

3b: Table #1 should be titled "Detailed characteristics and availability ... " as it covers "availability" as well.

*RESPONSE: Agreed and added to the caption of (now) Table 2 (page 13, line 16).*

3c: some columns in Table #1 are closely related to "fitness for use" expanded on in section #6 and alludes to at other points in the paper, so perhaps highlight them as such in the table and/or more explicitly refer back to them in the text (perhaps both when first introduced and in sections #5 and #6). This would help tie the two parts of the paper together (i.e., the "review" and the "fitness of use" sections).

*RESPONSE: We appreciate this suggestion, and added more references to this table in Section 5 (e.g., page 18, line 40; page 19, line 2 or lines 8/9). We prefer not to highlight more in the table itself to avoid cluttering in an already complex presentation.*

4: I agree that HYDE is a very interesting data set and perhaps of import to the readers of the journal but these data are unique historically and also are available at a fairly crude level compared to all other gridded data products. That is, is this product sufficiently different to include as say supplemental material rather than list with the others?

*RESPONSE: We agree with this notion that HYDE is very different in terms of historical context. However, after discussion with the editorial team it became clear that HYDE has been a rather prominent dataset used by the Earth Science community despite the coarseness and other aspects. Thus, we believe it is a valuable contrast between HYDE and the "newer generation" data products to highlight to the community that there are more options depending on the target application of interest.*

5: I wondered if the ancillary data section (#4.1) and Figure1 should come after section 4.2. Perhaps a relatively minor issue.

*RESPONSE: This comment is in line with Reviewer 1's suggestions (see response above) and we restructured the manuscript accordingly. This section (now Section 3.2, page 8, line 34) comes after the methods section (now Section 3.1, page 7, line 4).*

5b: Figure #1 has a lot of information that might be better explained. Ditto Figure #2.

*RESPONSE: We agree. We revised the figure itself (now Figure 2, page 9) to make it more legible (see also the comments made by Reviewer 1 and our response) and revised the figure caption (page 9, lines 9-16) as well as the text describing it (e.g., page 9, lines 5-7). Specifically, we put more emphasis on pointing out the temporal resolution and temporal mismatches between ancillary variables and population data, as well as explained the context of availability and quality of satellite data.*

*Similarly, we revised and redesigned the methods figure (now Figure 1, page 7), improved the text description and the figure caption (page 7, lines 20-24) to fit it better into the main text and included direct links to the different parts (page 7, line 9 to page 8, line 23).*

6: I know the readership of the journal is likely to be very sophisticated in this area but I still think that a glossary of key terms is necessary and should be presented early in the paper. Maybe I am thinking about my role as an instructor but I suspect in assigning this reading I would have to prepare the novice/intermediate user to several key terms.

*RESPONSE: We added a short Glossary at the beginning of the paper (page 2, lines 19-40) before the Introduction section) including some key terms that we think may be important for the novice reader to clarify.*

7: I very much liked Figures 3 and 4 and their brief description.

*RESPONSE: Thank you.*

8: I wondered if lines 11-15 could be bolded/italicized or perhaps just a separate paragraph to give emphasis to the interrelatedness of the determinants of fitness for use.

*RESPONSE: We changed some key words in these sentences to bold/italized style to put more emphasis on them – guiding the structure of the subsections that follow (page 18, lines 11-13).*

9: Fitness of use #2 focuses on urban population analysis. This seemed like an opportunity to link to Table 4 at www.popgrid.org/compare-data on "Global and Continental Urban Extent / Settlement Layers: Summary Characteristics."

***RESPONSE***: *We added a reference to two examples "(e.g., in GHS-POP and GRUMP)" to this paragraph rather than link to the online table 4 with a non-permanent link (page 21, line 30).*

10: Not sure there is, but if there was, an expansion of the theme "How have these data sets been used previously" would be useful. Some good examples, just wondering about classic papers or studies that could be used as exemplars.

***RESPONSE:*** *On page 4 lines 21-29 we list several applications in which gridded population data have been used. We slightly expanded and added several key applications but prefer not to add too many additional examples to avoid overwhelming the reader and excessive citation. We also maintain a footnote (Footnote 1, page 4) explaining that there are hundreds of applications but that this article is not focusing on a review of applications. Additionally, in the final Fitness-for-use section (page 22, lines 12-16) under "(6)* **How have these data sets been used previously?***" we link to the data producers' websites' list of citations.*

11: I recognize that there are pieces of the next comment scattered in the paper but for the user I wondered if more could be made of time-constant geographies and issues related to the embeddedness of units of analysis and scale. Some readers will be interested in trajectories of place as well as multilevel modeling (perhaps) but the number of temporal data collection points and the embeddedness of levels is not always made explicit. I agree that this can be complex but alluding to these issues in the context of research questions can help in the fitness of use.

***RESPONSE:*** *We agree that this is a complex issue but also appreciate the importance. In addition to the places we already mentioned related aspects, we included more text in Section 5.1 (page19, lines 9-13) to further strengthen this important issue as it relates to questions of use of the data products:*

*"Creating equivalencies over time of finely resolved census units is challenging even in vector format; this problem is not necessarily abated when transforming vector data to grids. Differences in embeddedness of the population grid cells within census boundaries (when the census units are intrinsically larger than the average grid-cell size, also has implications for subsequent analysis using e.g., multi-level models over large areas and can become even more problematic if the census boundary – grid cell relationship changes over time thus impeding the creation of reliable population trajectories of place."*

12: While not the main purpose I wasn't sure about all the referencing to social media. I wonder if the trap of thinking about the future has compelled the authors to discuss this but it all reads fairly superficially. At least to me. I do think an expanded section on data challenges (e.g., non-representative samples) and fast changing data environments (social network data, real-time data) are worth discussing but if so, then in more depth.

***RESPONSE:*** *The role of social media was discussed among co-authors and the team agreed that such novel data streams will become more important in future efforts also related to e.g., POPGRID activities. Thus, we only mention social media in the second-to-last sentence of the manuscript in future outlooks (page 23, line 2) and prefer not to discuss them in detail as they are currently not used in those data production activities.*

13: Minor–gridded datasets are not particularly recent and they are part of the history and formation of remote sensing, raster GIS, and map algebra tools/perspectives in the spatial sciences (in disciplines spanning the environmental, geographic, and social sciences). An early citation to time-constant geographies and socioeconomic applications in GIS that focused on grid-based data products include David Martin (1991, 1995), now at Southampton.

***RESPONSE***: *We added Martin and Bracken (1991) and Martin (1996) to the references.*

[revised manuscript text omitted]

**2015 Estimated Population (number of persons)**

| ☐ 0–1 | 🟧 5–25 | 🟥 250–1000 | 🟥 5000–25000 |
|---|---|---|---|
| 🟨 1–5 | 🟧 25–250 | 🟥 1000–5000 | 🟥 >25000 |

**ESACCI Landcover**

| 🟨 Cropland (rainfed) | 🟩 Mosaic vegetation | 🟧 Grassland |
|---|---|---|
| 🟩 Herbaceous cover | 🟩 Tree cover | 🟥 Urban |
| 🟩 Tree or shrub cover | 🟫 Shrubland | |

**OpenStreetMap Roads**

motorway — secondary — trunk

**WDPA Protected Areas**

🟩 Protected Areas

**Elevation (meters)**

| 🟫 <1800 | 🟫 1851–1900 | 🟫 1951–2000 | 🟨 >2051 |
|---|---|---|---|
| 🟫 1801–1850 | 🟫 1901–1950 | 🟫 2001–2050 | |

**WorldPop 2014 Population (persons per grid cell)**

[revised manuscript text omitted]

---

## Author Comment (AC2) · 21 Jul 2019

Dear Editor, dear Reviewers,

Please find the revised manuscript "The spatial allocation of population: A review of large-scale gridded population data products and their fitness for use" (revised title).

We addressed all comments made by the two reviewers point by point. In the uploaded file (pdf) you will find our responses to reviewers' comments first (pp. 1-7), followed by the revised manuscript. In our response, we included the page and line numbers for the revisions made in the manuscript to facilitate identifying the changes.

We would like to express our gratitude to Tracy Kugler and Stephen Matthews for their thoughtful comments and constructive critiques and suggestions. Addressing those comments and suggestions resulted in an improved manuscript. Your positive evaluation and support is greatly appreciated.

Please let us know if you need any additional clarifications.

Best regards,

Stefan Leyk (corresponding author) and the POPGRID collaborative.

Please also note the supplement to this comment:
https://www.earth-syst-sci-data-discuss.net/essd-2019-82/essd-2019-82-AC2-supplement.pdf